# Exploring the genomic resources of seven domestic Bactrian camel populations in China through restriction site-associated DNA sequencing

Chenmiao Liu, Huiling Chen, Xuejiao Yang, Chengdong Zhang, Zhanjun Ren *

College of Animal Science and Technology, Northwest A&F University, Yangling, shaanxi, China

* Renzhanjun@nwafu.edu.cn

**Data Availability Statement:** The datasets generated for this study are available under the NCBI Bioproject PRJNA522647, Biosample SAMN10948548-SAMN10948594.

## Abstract

The domestic Bactrian camel is a valuable livestock resource in arid desert areas. Therefore, it is essential to understand the roles of important genes responsible for its characteristics. We used restriction site-associated DNA sequencing (RAD-seq) to detect single nucleotide polymorphism (SNP) markers in seven domestic Bactrian camel populations. In total, 482,786 SNPs were genotyped. The pool of all remaining others were selected as the reference population, and the Nanjiang, Sunite, Alashan, Dongjiang, Beijiang, Qinghai, and Hexi camels were the target populations for selection signature analysis. We obtained 603, 494, 622, 624, 444, 588, and 762 selected genes, respectively, from members of the seven target populations. Gene Ontology classifications and Kyoto Encyclopedia of Genes and Genomes enrichment analyses were performed, and the functions of these genes were further studied using Genecards to identify genes potentially related to the unique characteristics of the camel population, such as heat resistance and stress resistance. Across all populations, cellular process, single-organism process, and metabolic process were the most abundant biological process subcategories, whereas cell, cell part, and organelle were the most abundant cellular component subcategories. Binding and catalytic activity represented the main molecular functions. The selected genes in Alashan camels were mainly enriched in ubiquitin mediated proteolysis pathways, the selected genes in Beijiang camels were mainly enriched in MAPK signaling pathways, the selected genes in Dongjiang camels were mainly enriched in RNA transport pathways, the selected genes in Hexi camels were mainly enriched in endocytosis pathways, the selected genes in Nanjiang camels were mainly enriched in insulin signaling pathways, while the selected genes in Qinghai camels were mainly enriched in focal adhesion pathways; these selected genes in Sunite camels were mainly enriched in ribosome pathways. We also found that Nanjiang (*HSPA4L* and *INTU*), and Alashan camels (*INO80E*) harbored genes related to the environment and characteristics. These findings provide useful insights into the genes related to the unique characteristics of domestic Bactrian camels in China, and a basis for genomic resource development in this species.

**Funding:** This study was supported by the National Natural Science Foundation of China, 31172178. The funders had no role in study design, data collection and analysis, decision to publish, or preparation of the manuscript.

**Competing interests:** The authors have declared that no competing interests exist.

## Introduction

The domestic Bactrian camel is an essential livestock resource in China, serving as an important means of transportation for cultural exchanges between the East and the West [1]. These camels are also the source of various livestock products. For example, camel hair is a good textile material containing natural protein fiber [2]. Camel milk is rich in nutrition and has hypoglycemic and anticancer activities [3,4]. Camel meat has high moisture and protein levels, as well as low fat and cholesterol levels, and has been shown to have medicinal properties, suggesting that it may have applications in the treatments of some diseases [5–7].

Seven domestic Bactrian camel populations come from different regions of China. Nanjiang camels have stronger heat resistance than the pool of all remaining others, Alashan camels form stronger anti-ultraviolet physiological functions than the pool of all remaining others [8,9]. Thus, the pool of all remaining others can be used as a reference group for exploration of the genomic resources of their species. At present, some studies have been conducted on domestic Bactrian camels. Ming et al. [10] sequenced the whole genome of 128 camels across Asia and revealed origin and migration of domestic Bactrian camels. Additionally, Liu et al. [11] used single nucleotide polymorphisms (SNPs) to study the genetic diversity and genetic structure of seven domestic Bactrian camel populations; our study clarified the phylogenetic relationships of these populations, laying a foundation for the protection of their biodiversity.

RAD-seq is a simplified genome sequencing technology based on whole-genome restriction sites developed on the basis of next-generation sequencing. With the establishment of high-throughput sequencing technology and bioinformatics technology, RAD-seq analysis has become increasingly refined, and has been applied to many organisms [12]. For example, Li et al. [13] used RAD-seq for genome SNP typing of 618 sows to accurately predict of genome breeding value and evaluate the accuracy of breeding value prediction. Moreover, Wang et al. [14] successfully employed RAD-seq to explore genome-wide SNPs among six breeds of Sichuan cattle.

In this study, we explored the genome-wide SNP markers of seven domestic Bactrian camel populations by RAD-seq. Our aim was to identify genes that might be related to the unique characteristics of these camels, such as heat and stress resistance. Our results established interesting targets for subsequent genomics research of the domestic Bactrian camel, and may facilitate genetic association analysis of economically important traits of this species.

## Materials and methods

### Animals and sampling sites

Blood sampled from domestic Bactrian agricultural camels were obtained from seven sites in Inner Mongolia, Gansu, Qinghai, and Xinjiang (Table 1, Fig 1). A total of 47 venous blood samples were collected (5 ml each). Miannaining, a compound preparation of xylazine and dihydroetorphine hydrochloride (with a specification of 2 ml per tube) purchased from the Institute of Quartermaster University of the Chinese People's Liberation Army, was used to anaesthetize the animals during the sampling process [15]. The application amount of camel was 2.5–3.5ml/100kg, and each sample was derived from a different family; there was no kinship among individuals [16]. All experimental protocols from this study were approved by the Institutional Animal Care and Use Committee of the College of Animal Science and Technology, Northwest A & F University (Shaanxi, China; approval no. DNX20170604).

### DNA extraction, library construction, and Illumina sequencing

DNA samples were extracted following a standard phenol-chloroform extraction procedure before being diluted to 20 ng/μL [17]. Then, the quality and quantity of DNA were evaluated

**Table 1. Information for domestic Bactrian camels in the seven sampled sites of China.**

| Code | Population | Location | Latitude | Longitude | Number |
|------|-----------|----------|----------|-----------|--------|
| NJ | Nanjiang camel | Wensu county, Xinjiang | 41°29′ | 80°24′ | 7 |
| BJ | Beijiang camel | Qinghe county, Xinjiang | 46°71′ | 90°37′ | 7 |
| DJ | Dongjiang camel | Mulei county, Xinjiang | 43°80′ | 90°34′ | 7 |
| HX | Hexi camel | Yongchang county, Gansu | 38°25′ | 101°97′ | 5 |
| QH | Qinghai camel | Mohe, Qinghai | 36°77′ | 99°05′ | 7 |
| ALS | Alashan camel | Alashan Left Banner, Inner Mongolia | 38°85′ | 105°68′ | 7 |
| SNT | Sunite camel | Sunite Right Banner, Inner Mongolia | 42°47′ | 112°95′ | 7 |
| Total | | | | | 47 |

using an Epoch spectrophotometer (BioTek, Winooski, VT, USA), a Qubit 3.0 fluorometer (Thermo Fisher Scientific, Waltham, MA, USA), and agarose gel electrophoresis [18].

RAD-seq libraries were constructed in accordance with a modified protocol [19]. Genomic DNA (0.1–1 µg; from a single sample or pooled samples) was digested with EcoRI (New England Biolabs), and P1 adapters were connected at the cutting site. The samples were then pooled, randomly sheared, and size-selected. After P2 adapters were added, DNA fragments ranging from 300 to 700 bp were used to construct the sequencing libraries. Finally, the samples were sequenced on the Illumina HiSeq 3000 platform (Illumina, San Diego, CA, USA) using 100-bp paired-end reads.

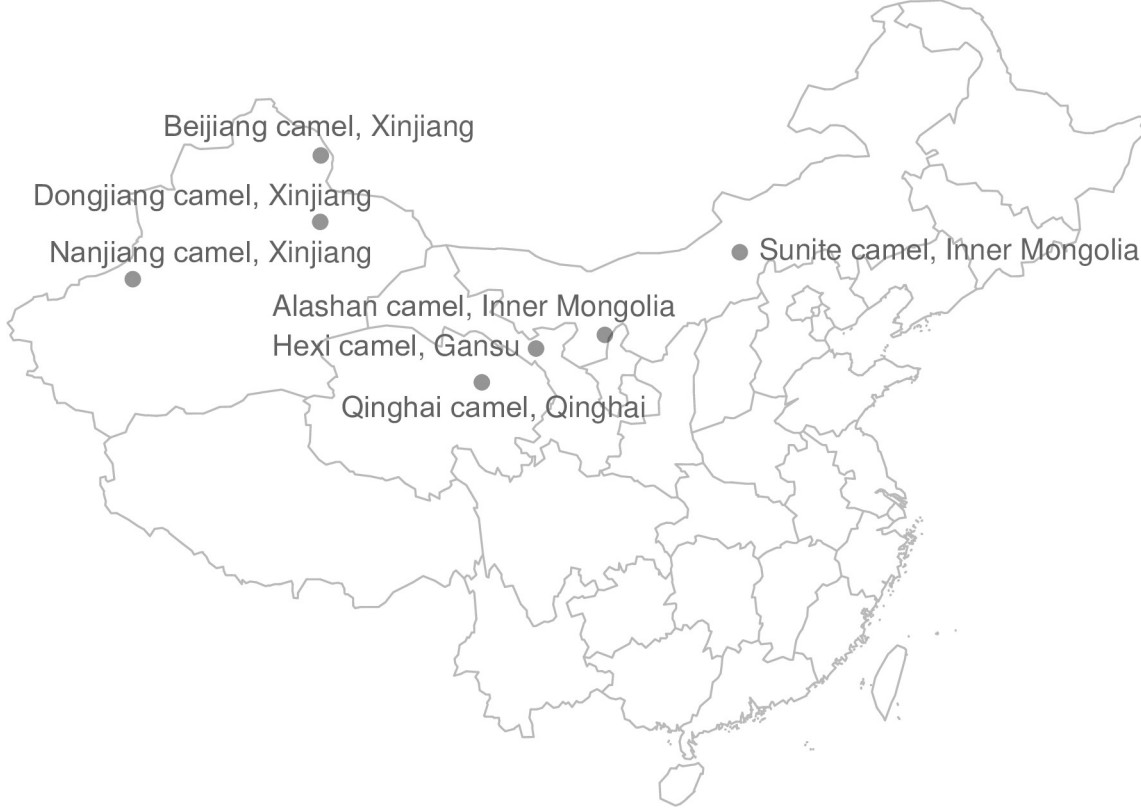

**Fig 1. A geographical distribution map of seven domestic Bactrian camel populations in China.** ALS, Alashan camels; BJ, Beijiang camels; DJ, Dongjiang camels; HX, Hexi camels; NJ, Nanjiang camels; QH, Qinghai camels; SNT, Sunite camels.

## Quality control, read mapping, and SNP calling

The fastp tool (v0.19.5) was used to filter the generated reads [20]. Raw reads were processed into high-quality clean reads using three strict filtering standards, as follows: (1) removing reads with greater than or equal to 10% unidentified nucleotides (N), (2) removing reads with greater than 50% of bases having Phred quality scores of less than or equal to 20, and (3) removing reads aligned to the barcode adapter.

Burrows-Wheeler Aligner software was used to align the clean reads of each sample with the reference genome (https://www.ncbi.nlm.nih.gov/genome/10741), with settings of 'mem 4 -k 32 -M', where -k is the minimum seed length and -M is an option used to mark shorter split alignment hits as secondary alignments [21]. GATK's Unified Genotyper was used to conduct variant calling on all samples [22], and GATK Variant Filtration, with proper standards, was used to filter SNPs (-Window 4, -filter "QD < 2.0 || FS > 60.0 || MQ <40.0", -G_filter "GQ < 20") [23]. Finally, the obtained SNPs were filtered with VCFtools (https://github.com/vcftools/vcftools) for further analysis with Minor Allele Frequency (MAF) > 0.05 and proportion of missing genotyping data < 25% as parameters. This data file was then used in subsequent analyses [24].

## Analysis of the selection signatures

Selection signature detection was performed using 50-kb windows and 25-kb steps for sliding the π value (nucleotide diversity) and for $F_{ST}$ (population differentiation index) distribution calculation. The -log10 transform of Nei's π was used to select the lower end of the diversity windows, with these parameters then quantified using in-house PERL scripts (S1 File). The top 5% of the $F_{ST}$ and π values were selected as candidate regions [25,26]. All related graphs were drawn using R scripts [27].

## Gene Ontology (GO) enrichment analysis and Kyoto Encyclopedia of Genes and Genomes (KEGG) pathway enrichment analysis

To further systematically elucidate their complex biological functions, the selected genes from Nanjiang, Sunite, Alashan, Dongjiang, Beijiang, Qinghai, and Hexi camels were mapped against both GO and KEGG databases. WEGO software was used for GO enrichment analysis, and the gene number of each term was calculated [28]. KOBAS software and the KEGG database (http://www.genome.jp/kegg/) were used to test the statistical enrichment of the selected genes [29]. The calculated *p* values were subjected to false discovery rate (FDR) correction, applying an FDR threshold less than or equal to 0.05. Pathways meeting this condition were defined as significantly enriched pathways.

# Results

## RAD-seq sequencing results and data filtering

A total of 131 GB of raw data and 893,672,274 reads from seven populations were obtained by Illumina sequencing. After quality filtering, 129 GB of clean data and 882,097,022 clean reads remained. The average clean data for each sample was 2.75 GB, and an average of 19.01 million reads was obtained. The average effective rate was 98.57%. Q20 was higher than 94%, Q30 was higher than 87%, and the GC content was stable between 40.71% and 45.13% (S1 Table). Overall, the sequencing data showed high Phred quality.

## Analysis of the selection signatures

$F_{ST}$ and π were used to select the top 5% regions. Using the pool of all remaining others as the reference population and Nanjiang, Sunite, Alashan, Dongjiang, Beijiang, Qinghai, and Hexi camels as the target populations, 603, 494, 622, 624, 444, 588, and 762 selected genes, respectively, were obtained (Figs 2–8, S2–S8 Tables). The selected genes in the scanning regions were extracted, and their functions were predicted using Genecards. The analysis indicated that the Nanjiang camels (*HSPA4L*, *INTU*), and Alashan camels (*INO80E*) may harbor genes related to the environment and their unique characteristics.

## Sequence annotation and enrichment analysis

GO classification was carried out on the selected genes. Across all populations, cellular process (GO:0009987), single-organism process (GO:0044699), and metabolic process (GO:0008152)

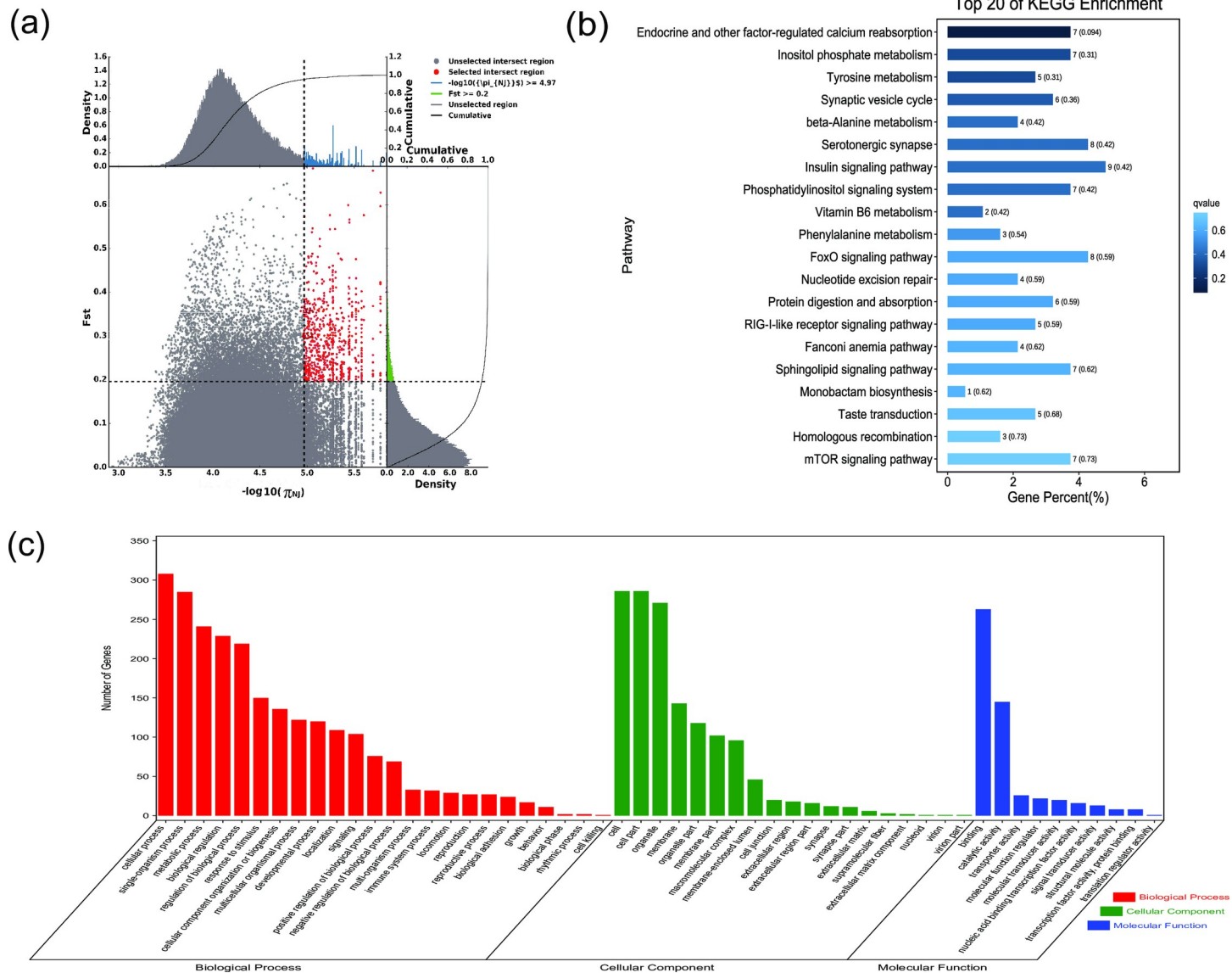

**Fig 2. GO classification and KEGG enrichment of the selected genes of Nanjiang camels.** (a) With the pool of all remaining others as the reference population and Nanjiang camels as the target population, 603 selected genes were obtained. (b) KEGG enrichment of 603 selected genes of Nanjiang camels. (c) GO classification of 603 selected genes of Nanjiang camels.

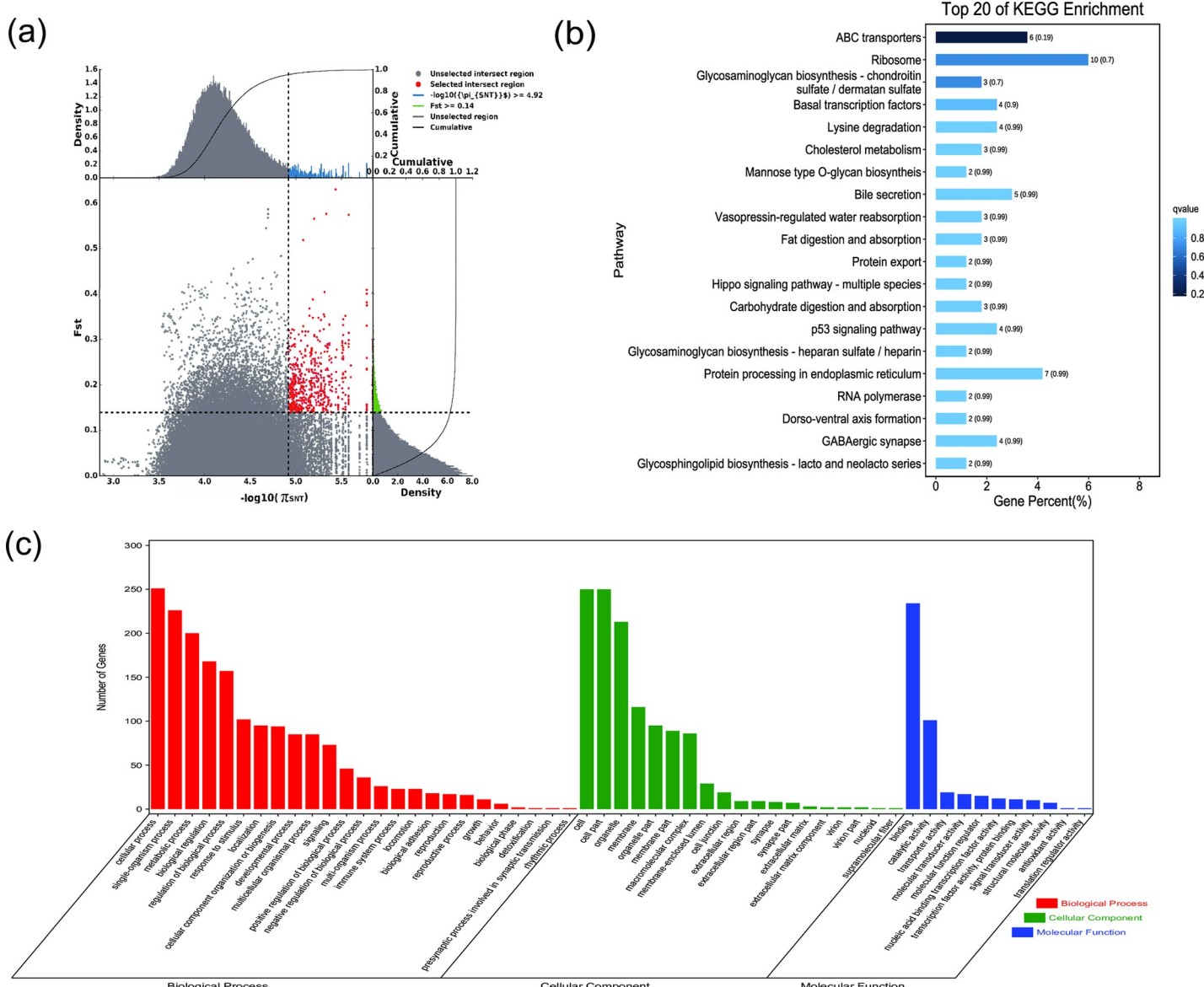

**Fig 3. GO classification and KEGG enrichment of the selected genes of Sunite camels.** (a) With the pool of all remaining others as the reference population and Sunite camels as the target population, 494 selected genes were obtained. (b) KEGG enrichment of 494 selected genes of Sunite camels. (c) GO classification of 494 selected genes of Sunite camels.

were the most abundant biological process subcategories. Cell (GO:0005623), cell part (GO:0044464), and organelle (GO:0043226) were the most abundant cellular component categories. Binding (GO:0005488) and catalytic activity (GO:0003824) represented the main molecular functions (Figs 2–8, S9–S15 Tables).

KEGG enrichment analysis was carried out on the selected genes from seven domestic Bactrian camel populations. In Alashan camels, the selected genes were mainly enriched in ubiquitin mediated proteolysis pathways. In Beijiang camels, the selected genes were mainly enriched in MAPK signaling pathways. In Dongjiang camels, the selected genes were mainly enriched in RNA transport pathways. In Hexi camels, the selected genes were mainly enriched in endocytosis pathways. In Nanjiang camels, the selected genes were mainly enriched in insulin

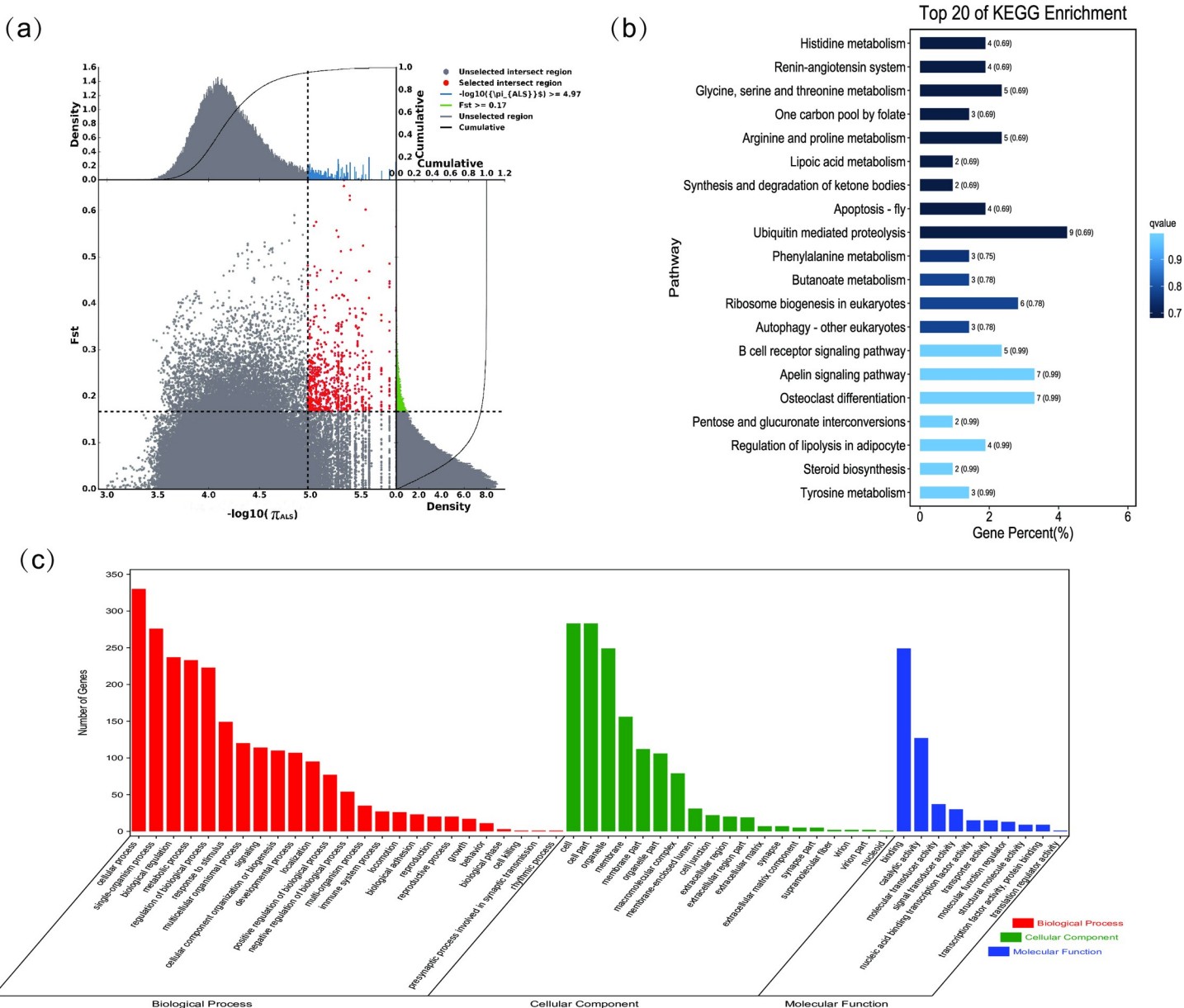

**Fig 4. GO classification and KEGG enrichment of the selected genes of Alashan camels.** (a) With the pool of all remaining others as the reference population and Alashan camels as the target population, 622 selected genes were obtained. (b) KEGG enrichment of 622 selected genes of Alashan camels. (c) GO classification of 622 selected genes of Alashan camels.

signaling pathways. In Qinghai camels, the selected genes were mainly enriched in focal adhesion pathways. In Sunite camels, the selected genes were mainly enriched in ribosome pathways (Figs 2–8, S16–S22 Tables).

## Discussion

In this study, we used RAD-seq to detect 482,786 SNP markers from seven domestic Bactrian camel populations. The pool of all remaining others were used as the reference population for selection signature analysis. GO classification and KEGG enrichment analysis were performed

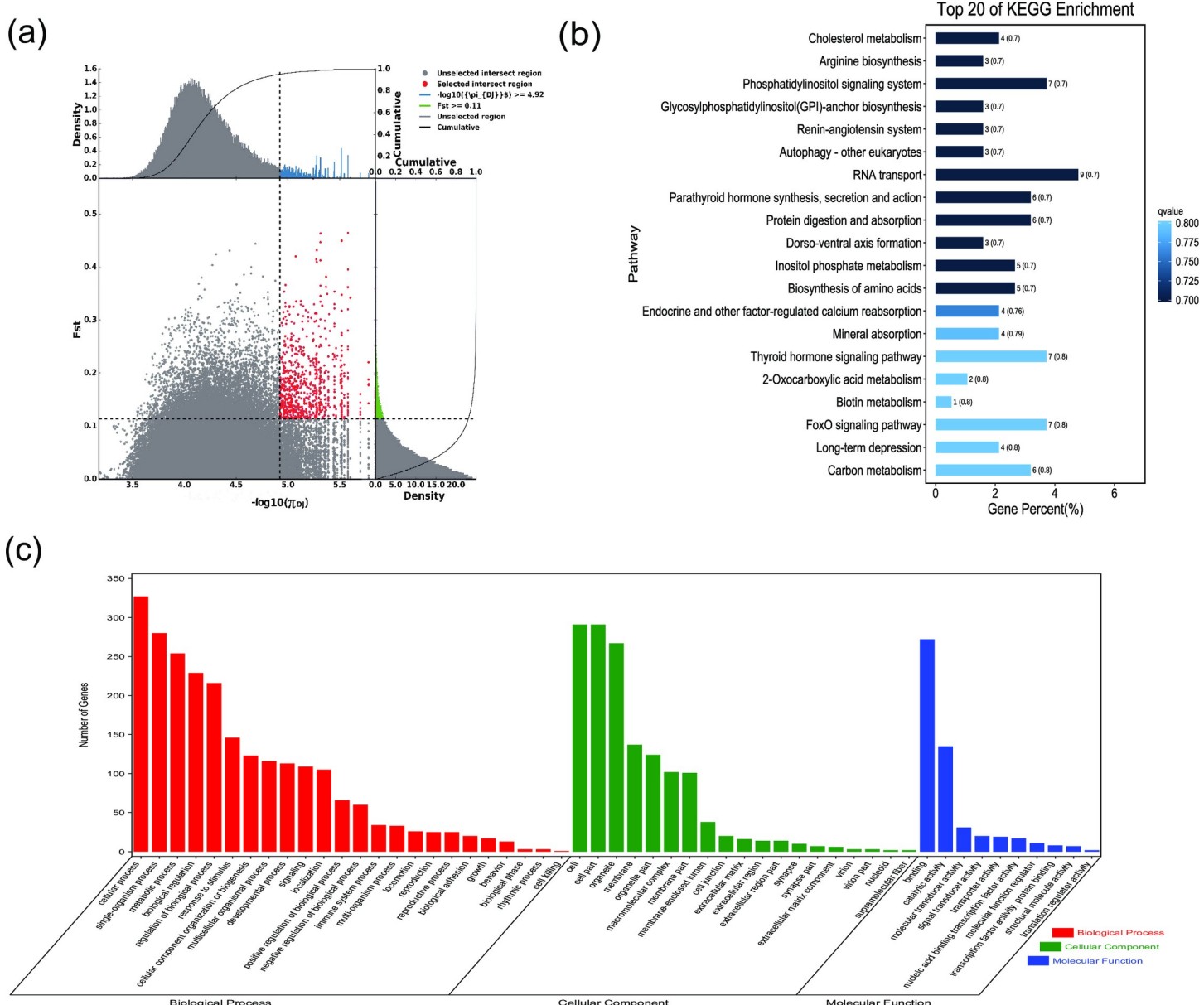

**Fig 5. GO classification and KEGG enrichment of the selected genes of Dongjiang camels.** (a) With the pool of all remaining others as the reference population and Dongjiang camels as the target population, 624 selected genes were obtained. (b) KEGG enrichment of 624 selected genes of Dongjiang camels. (c) GO classification of 624 selected genes of Dongjiang camels.

on the identified selected genes, and Genecards was used to further study their functions. It showed that Nanjiang, and Alashan camels may harbor genes related to the environment and the unique characteristics of these camels. Overall, our findings established interesting targets for genetic association analysis of important traits in domestic Bactrian camel populations in China, and provided a basis for additional genome studies in this species [30–32].

Wensu County, the home of Nanjiang camels, is close to the Taklimakan Desert. Nanjiang camels have long lived in an environment with abundant sunlight, little rain, strong winds, and copious amounts of sand [33]. Thus, these camels tend to have strong heat resistance [8]. *HSPA4L* (encoding heat shock protein family A member 4 like) is a candidate gene related to

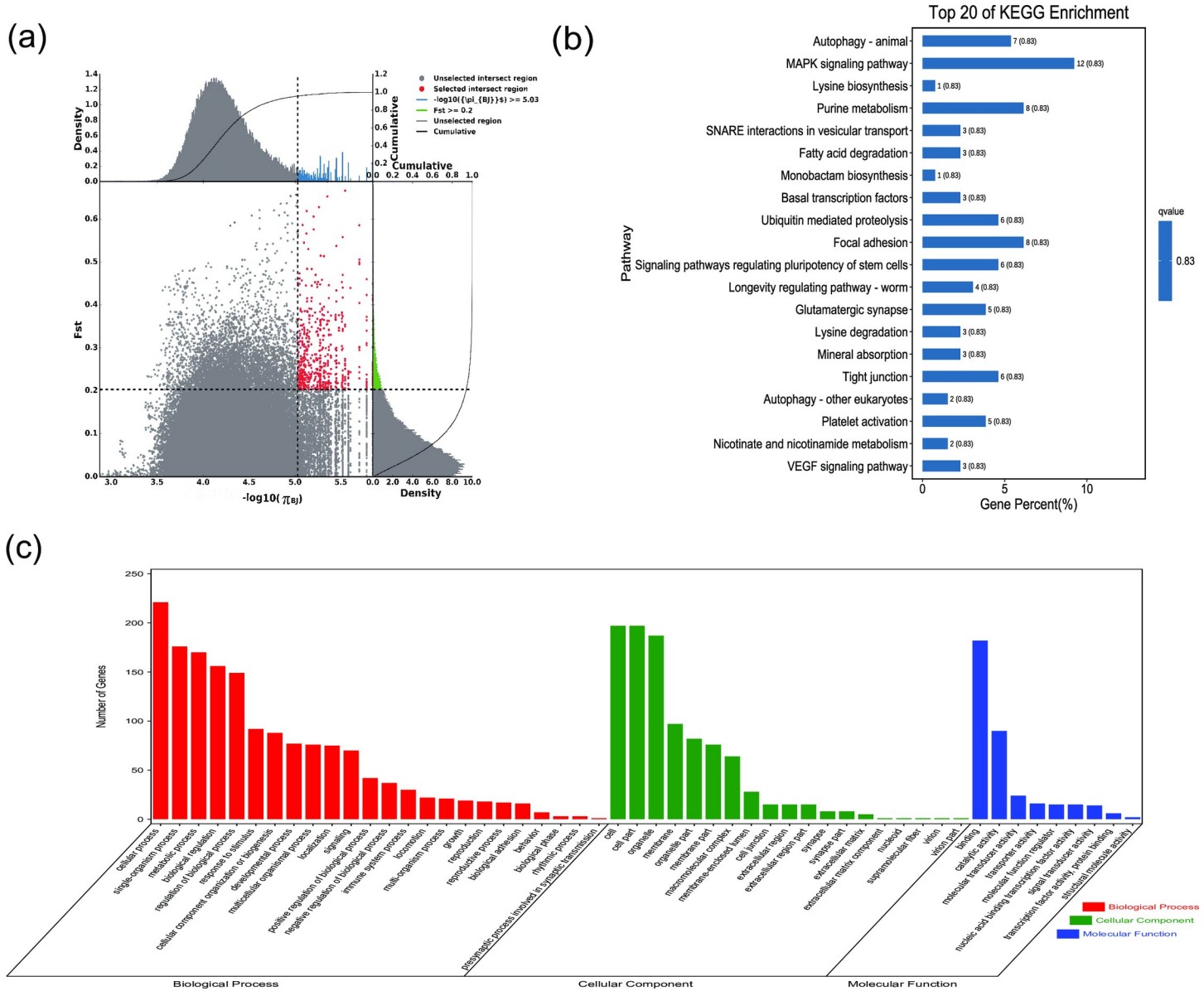

**Fig 6. GO classification and KEGG enrichment of the selected genes of Beijiang camels.** (a) With the pool of all remaining others as the reference population and Beijiang camels as the target population, 444 selected genes were obtained. (b) KEGG enrichment of 444 selected genes of Beijiang camels. (c) GO classification of 444 selected genes of Beijiang camels.

heat resistance [30] and cellular response to heat stress. Thus, *HSPA4L* may be an interesting target for studying heat resistance in Nanjiang camels. Additionally, these camels have 3–5-cm-long eyelashes, with eyelash length determined by the duration of their hair follicle growth period [34]. The Inturned planar cell polarity protein (encoded by *INTU*) is involved in hair follicle morphogenesis [35]. Therefore, studies of *INTU* may provide insights into the distinct eyelash length of Nanjiang camels. Alashan camels originate from the Alashan Plateau, in western Inner Mongolia. The Alashan Plateau is subjected to strong ultraviolet rays, and Alashan camels have evolved anti-ultraviolet physiological functions in response to this prolonged exposure [9]. Because *INO80E* (INO80 complex subunit E) is involved in UV-damage

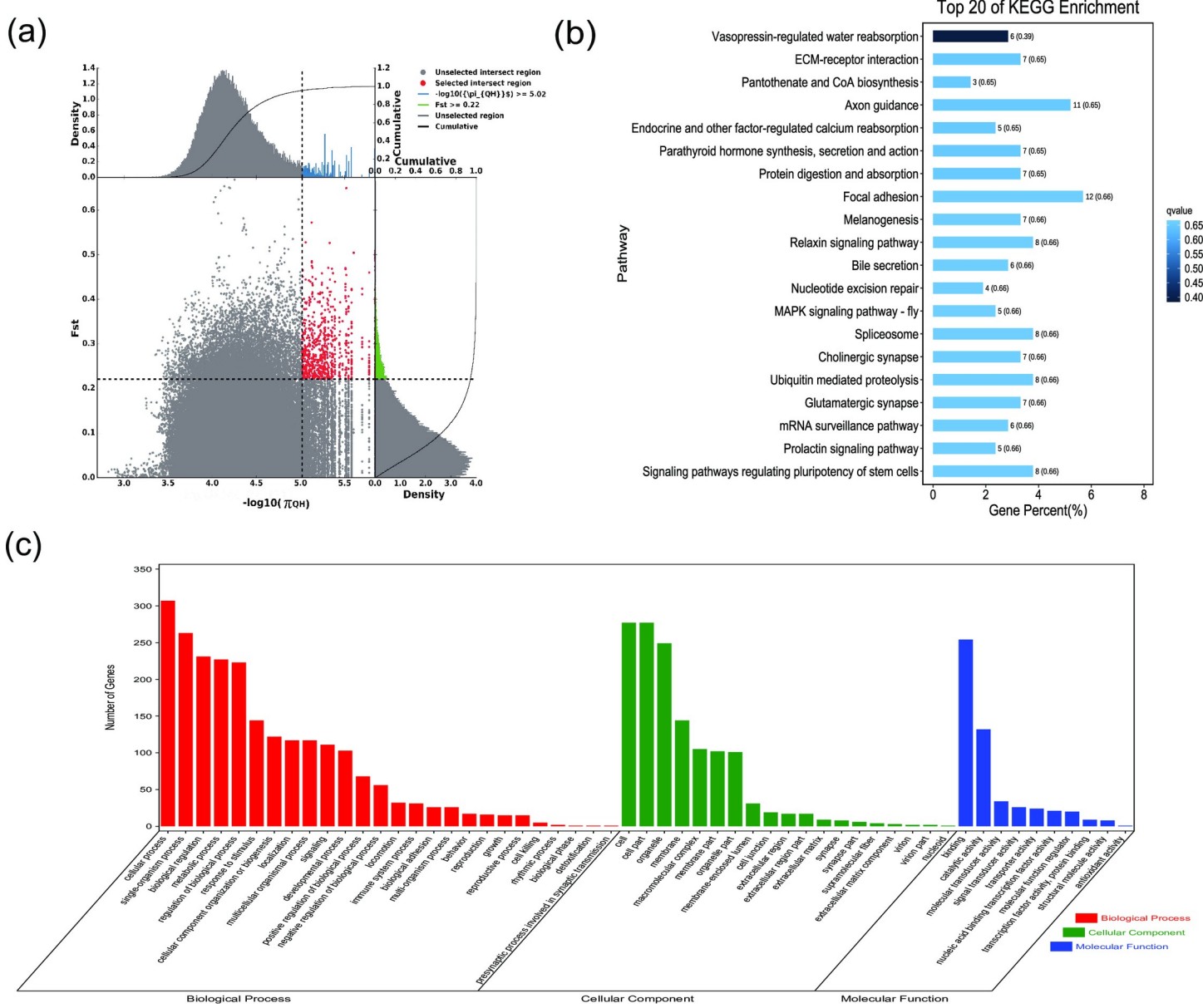

**Fig 7. GO classification and KEGG enrichment of the selected genes of Qinghai camels.** (a) With the pool of all remaining others as the reference population and Qinghai camels as the target population, 588 selected genes were obtained. (b) KEGG enrichment of 588 selected genes of Qinghai camels. (c) GO classification of 588 selected genes of Qinghai camels.

excision repair [36], this gene may have applications in future studies of anti-ultraviolet radiation functions in these camel populations.

In summary, the SNP loci of seven domestic Bactrian camel populations in China were identified at the whole-genome level using RAD-seq. Through selection signature analysis, we found that Nanjiang, and Alashan camels may have genes related to the environment and the unique characteristics of these camels. These findings have established a basis for further mining the genomic resources of domestic Bactrian camels in China, which will help us better protect and develop the genetic resources of the species. In future studies, we will actively seek cooperation with other institutions to conduct a more in-depth exploration of genes related to the characteristics of this species, furthering our understanding of their genomic resources.

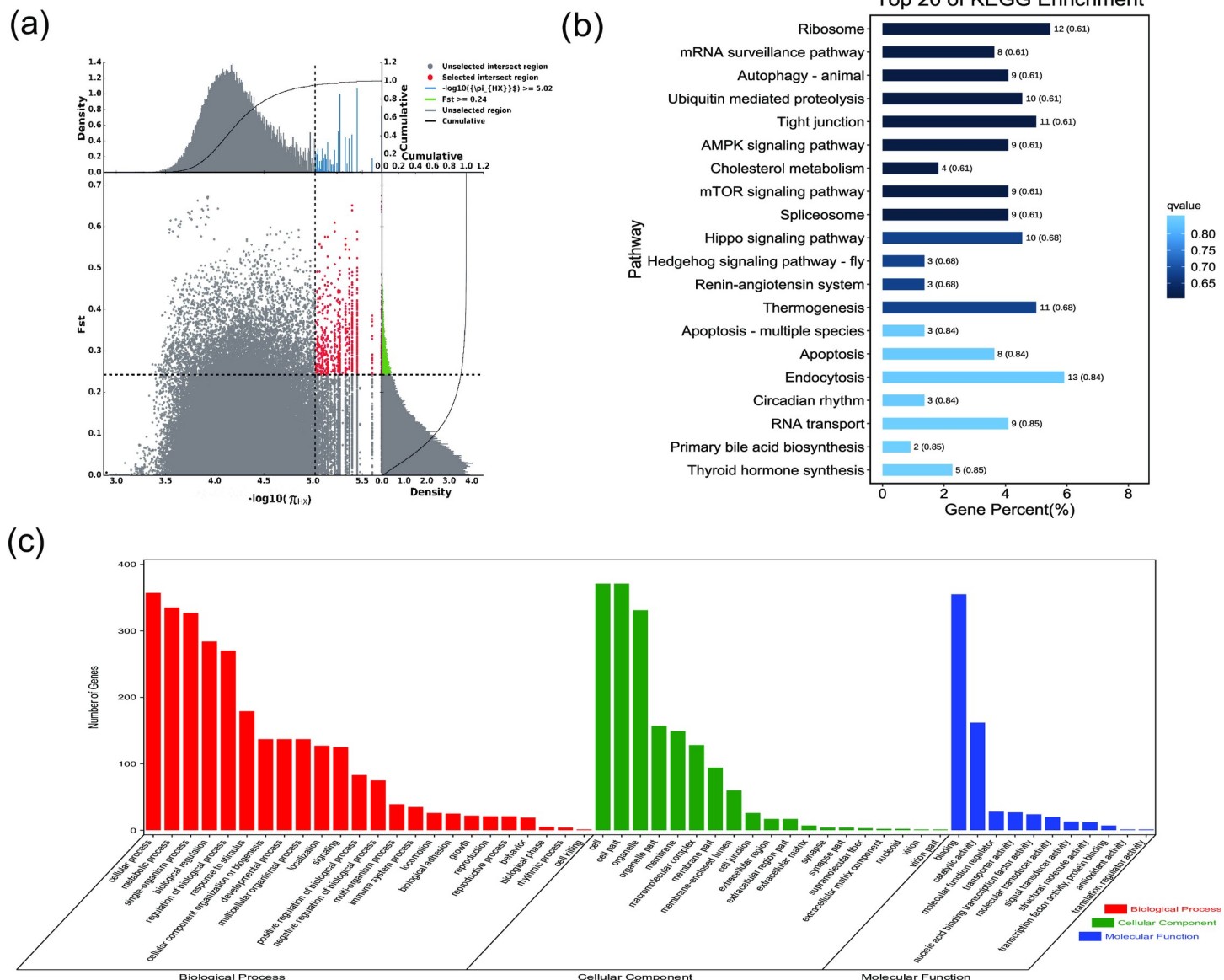

**Fig 8. GO classification and KEGG enrichment of the selected genes of Hexi camels.** (a) With the pool of all remaining others as the reference population and Hexi camels as the target population, 762 selected genes were obtained. (b) KEGG enrichment of 762 selected genes of Hexi camels. (c) GO classification of 762 selected genes of Hexi camels.

## Supporting information

**S1 Table. Sequencing results and quality filtering of reads.**
(DOCX)

**S2 Table. The pool of all remaining others_vs_NJ_0.05_fst_pi_sig_intersect.Annot.**
(XLSX)

**S3 Table. The pool of all remaining others_vs_SNT_0.05_fst_pi_sig_intersect.Annot.**
(XLSX)

**S4 Table. The pool of all remaining others_vs_ALS_0.05_fst_pi_sig_intersect.Annot.**
(XLSX)

**S5 Table. The pool of all remaining others_vs_DJ_0.05_fst_pi_sig_intersect.Annot.**
(XLSX)

**S6 Table. The pool of all remaining others_vs_BJ_0.05_fst_pi_sig_intersect.Annot.**
(XLSX)

**S7 Table. The pool of all remaining others_vs_QH_0.05_fst_pi_sig_intersect.Annot.**
(XLSX)

**S8 Table. The pool of all remaining others_vs_HX_0.05_fst_pi_sig_intersect.Annot.**
(XLSX)

**S9 Table. Gene Ontology (GO) classification of the selected genes in Sunite camel.**
(XLSX)

**S10 Table. Gene Ontology (GO) classification of the selected genes in Alashan camel.**
(XLSX)

**S11 Table. Gene Ontology (GO) classification of the selected genes in Dongjiang camel.**
(XLSX)

**S12 Table. Gene Ontology (GO) classification of the selected genes in Beijiang camel.**
(XLSX)

**S13 Table. Gene Ontology (GO) classification of the selected genes in Qinghai camel.**
(XLSX)

**S14 Table. Gene Ontology (GO) classification of the selected genes in Nanjiang camel.**
(XLSX)

**S15 Table. Gene Ontology (GO) classification of the selected genes in Hexi camel.**
(XLSX)

**S16 Table. KEGG enrichment of the selected genes in Sunite camel.**
(XLSX)

**S17 Table. KEGG enrichment of the selected genes in Alashan camel.**
(XLSX)

**S18 Table. KEGG enrichment of the selected genes in Dongjiang camel.**
(XLSX)

**S19 Table. KEGG enrichment of the selected genes in Beijiang camel.**
(XLSX)

**S20 Table. KEGG enrichment of the selected genes in Qinghai camel.**
(XLSX)

**S21 Table. KEGG enrichment of the selected genes in Nanjiang camel.**
(XLSX)

**S22 Table. KEGG enrichment of the selected genes in Hexi camel.**
(XLSX)

**S1 File. The in-house PERL scripts.**
(TXT)

## Author Contributions

**Investigation:** Huiling Chen.

**Methodology:** Xuejiao Yang, Chengdong Zhang.

**Writing – original draft:** Chenmiao Liu.

**Writing – review & editing:** Zhanjun Ren.

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
