## [Decision Letter · Decision Letter 0]

26 Aug 2020

PONE-D-20-20388

Exploring the genom ic  resources of seven domestic Bactrian camel populations in China by RAD-seq

PLOS ONE

Dear Dr. Ren,

Thank you for submitting your manuscript to PLOS ONE. After careful consideration, we feel that it has merit but does not fully meet PLOS ONE’s publication criteria as it currently stands. Therefore, we invite you to submit a revised version of the manuscript that addresses the points raised during the review process.

We look forward to receiving your revised manuscript.

Kind regards,

Tzen-Yuh Chiang

Academic Editor

PLOS ONE

Journal Requirements:

2. In your Methods, please describe the exact protocol used for blood sampling, including the volume of the samples and whether any analgesia was used.

Reviewers' comments:

Reviewer's Responses to Questions

**Comments to the Author**

1. Is the manuscript technically sound, and do the data support the conclusions?

Reviewer #1: Yes

Reviewer #2: No

2. Has the statistical analysis been performed appropriately and rigorously? 

Reviewer #1: Yes

Reviewer #2: No

3. Have the authors made all data underlying the findings in their manuscript fully available?

Reviewer #1: Yes

Reviewer #2: No

4. Is the manuscript presented in an intelligible fashion and written in standard English?

Reviewer #1: No

Reviewer #2: No

5. Review Comments to the Author

Reviewer #1: In this manuscript, RAD-seq was used to detect SNP markers from seven distinct domestic Bactrian camel populations from China. SNPs were used for Fixation index (FST) and principal component (PCA) analysis, and involved genes were tested for possible selection, where their function was further studied. In my opinion, this manuscript is of importance to the scientific community and has great potential. Nevertheless, I have several suggestions and comments in which I believe it might improve the impact of it.

General comments and suggestions:

1. In general, I feel that connection between sentences is missing throughout the whole text in order to create a better flow while reading. See, for example, l.50 – 55. I would suggest to review the whole manuscript and create better links between sentences as well as sentence construction. The discussion is also very descriptive and lacks connection between sentences (more comments about the discussion below).

2. Some abbreviations for text comprehension have no explanation assuming all readers know what is relates to, for example l.59: MSY. I would suggest to review all abbreviations.

3. The authors wrote several times in the different manuscript sections that FST and PCA were used for testing “phylogenetic” distances from domestic Bactrian camels in different populations. But the authors did not do any phylogenetic analysis, they have just tested for population differentiation (Fst) and individual-to-individual (PCA) genetic distances. I suggest the authors change it to “genetically” instead.

4. At the beginning the taxonomic description of the domestic Bactrian camel is missing. Only in the line 59 it is stated the scientific name, but with no connection to the common name.

5. I don’t understand why the authors justify the use of Hexi camel population as “control population” for selection tests just because “these camels are genetically more distant to the other populations”, to then infer environmental pressures or production traits. I am not saying it is a bad choice, I am just commenting and calling attention to the fact that there is context information missing to understand the thought behind it. Why did the authors choose these populations specifically? What do they have in common and what do they have that differentiates them? Do all populations except for Hexi live under the same environmental conditions? Are the animals under different selective pressures on production traits on the control population compared to the others? It is always very interesting to assess genes under selection, but it is also very important to contextualize the results with the real situation, in this case, contextualize the results with the different camel populations the authors are using. I comment more on this in the “introduction” and “discussion” detailed suggestions where I suggest less descriptive text on the discussion and more information on these populations (maybe this information should start in the introduction section) to better understand the whole manuscript message.

Detailed comments and suggestions:

Abstract.

6. L15-17: “In this study, restriction site associated DNA sequencing (RAD-seq) was used to detect SNP markers of the seven domestic Bactrian camel populations.” There are not only seven Bactrian populations in China, so I would suggest to delete “the”. Also, it is “Restriction site-associated DNA sequencing” (hyphen missing).

7. L. 17 – 20: “The filtered SNPs were used for Fixation index (FST) analysis and principal component analysis (PCA) to find out the population which is phylogenetically distant from domestic Bactrian camels in other populations as the control population.” I suggest to rephrase this sentence.

Introduction.

8. L. 55 – 57: “In addition, the domestic Bactrian camel lives in hot and arid desert environment, which has potential ecological adaptation value.” Domestic Bactrian camels do not live exclusively in hot and arid, but also in cold and high altitudes. I would suggest to rephrase this sentence or to justify where specifically they live in these conditions – again, it would help if there would be already here more information on the different populations that were chosen in this study.

9. L. 58: delete “(2019)”.

10. L. 59: “Camelus bactrianus” should be in italic.

11. L. 58 – 66: The authors start by telling about this other study (in the 3rd person), and then on the l.63 they state “in the previous study, we used…”. I would suggest to be coherent and either write it in the 1st or 3rd person.

12. L. 70 - 71: “RAD-seq analysis is becoming more and more perfect” I suggest to rephrase “more and more perfect”.

13. L. 71 – 78: It is good that the authors state examples of studies that used the same technique in different species like eggplant or chicken, but I would suggest to state examples closer to camels or even within camelid species, that have used the same (RADseq) technique.

14. L. 82 – 87: Please add citations or links to “GO”, “KEGG” and “Genecards”.

15. Missing a more complete sentence in the end of the introduction with the aim and importance of this study as right now these are not very clearly stated.

Materials and Methods.

16. L. 123-128: substitute “readings” to “reads”.

17. The authors did SNP filtering, but did the authors do sample filtering (e.g. filtering samples out with missing call of a specific rate)?

18. L. 154: are the “internal PERL scripts” available to the scientific community, to promote transparency?

19. I think it would be interesting to see ADMIXTURE analysis in order to infer population structure and for calculating ancestry estimates.

Results.

20. L. 179: I suggest to stated with how many SNP the analysis were performed.

21. L. 186: The respective initials of the populations to the populations in the table description are not stated.

22. L. 191: please make reference to the figure number.

Discussion.

23. L. 264: I would suggest to change “In the study” to “In this study”.

24. L. 266: Fst in italic.

25. I really feel it is missing connection between the results and the camel populations history. I think the manuscript would improve greatly if the authors would change from a descriptive discussion to a more “story telling” format around the subject. This means that maybe there should be a bit more contextualization of these populations first (starting already in the introduction), and then relate to their findings, instead of just being very descriptive in major results. This would be very helpful to better understand their findings. This also connects to one of my comments on creating better flow between sentences.

26. L.284-285: Please elaborate on how this finding fills in the knowledge gap.

Reviewer #2: Manuscript PONE-D-20-20388 seemed to be designed, based on the same RAD-seq data reported before by the same group of authors (Liu CM, Chen HL, Ren ZJ, Zhang CD, Yang XJ. Population genetic analysis of the domestic Bactrian camel in China by RAD-seq. Ecology and evolution. 2019;34(4): 11232-11242. doi: 10.1002/ece3.5624.), to examine the population differentiation and then selection signals of the seven Chinese Bactrian camels. After reading their early article published in 2019, it is hard to see the value of publishing this manuscript with limited new result.

1. The manuscript is written in very poor English.

2. The data and major analyses including FST and principle component analyses were already included in the article published by the same authors.

3. The selection signal analyses based on the π value and FST values from the comparison of the Hexi camel population with the other six camel populations make almost no sense. Considering the very low number of samples and also genetic admixture of the Hexi camels with Dongjiang, Qinghai or Sunite camels as shown in the article of 2019 and also the genetic differentiation of the Qinghai camels at PC1 in this manuscript, it is not justified to treat Hexi camels as a control population. Therefore, the annotations of the what so called functional genes are not reliable at all.

4. Further detailed comments are shown in the reviewed manuscript using the annotation tools in the PDF format.

6. PLOS authors have the option to publish the peer review history of their article (what does this mean?). If published, this will include your full peer review and any attached files.

Reviewer #1: No

Reviewer #2: No

---

## [Author Response · Author response to Decision Letter 0]

15 Sep 2020

PONE-D-20-20388

Exploring the genomic resources of seven domestic Bactrian camel populations in China by restriction site-associated DNA sequencing

PLOS ONE

Dear Editor and Reviewers,

Thank you very much for the constructive comments made by the editor and reviewers to our manuscript “Exploring the genomic resources of seven domestic Bactrian camel populations in China by restriction site-associated DNA sequencing”. I have read all the comments carefully and answered each question one by one. Where appropriate we have also included the altered or additional text after this, shown in green font, as well as line numbers in the revised manuscript. We hope this is now suitable for publication in PLoS One. Once again, thank the editor and reviewers for their constructive comments. The main comments and our specific responses are detailed below.

Sincerely,

Chenmiao Liu

(On behalf of the co-authors)

Journal Requirements:

1.Please ensure that your manuscript meets PLOS ONE's style requirements, including those for file naming. 

Response: Thank you very much for your review.

Has been revised.

I ensure that my manuscript meets PLOS ONE's style requirements, including those for file naming.

2.In your Methods, please describe the exact protocol used for blood sampling, including the volume of the samples and whether any analgesia was used.

Response: Thank you very much for your review.

Has been revised.

P5L94-97.

In total, 47 venous blood samples were collected (5 mL each). Miannaining was used during the sampling process, and each sample was derived from a different family; there was no kinship among individuals. 

3.We suggest you thoroughly copyedit your manuscript for language usage, spelling, and grammar. 

Response: Thank you very much for your review.

I have copyedited my manuscript thoroughly for language usage, spelling, and grammar.

Reviewers' comments:

1.Is the manuscript technically sound, and do the data support the conclusions?

Response: Thank you very much for your review.

Here, we used restriction site-associated DNA sequencing (RAD-seq) to detect single nucleotide polymorphism (SNP) markers in seven domestic Bactrian camel populations. The identified SNPs were further evaluated using fixation index (FST) and principal component analysis (PCA). FST and PCA showed that Hexi camels were genetically distant from domestic Bactrian camels in other populations. Based on differences in geographical location, farming, and animal husbandry production areas/modes, with Hexi camels as the control population and Nanjiang, Sunite, Alashan, Dongjiang, Beijiang, and Qinghai camels as the selection populations for selection signature analysis. Gene Ontology (GO) classifications and Kyoto Encyclopedia of Genes and Genomes (KEGG) enrichment analyses were performed, and the functions of these genes were further studied by Genecards to identify the genes that may be related to the unique characteristics of the camel population, such as heat resistance and stress resistance. 

2.Has the statistical analysis been performed appropriately and rigorously?

Response: Thank you very much for your review.

In this study, we explored genome-wide SNP markers of seven domestic Bactrian camel populations by RAD-seq. The identified SNPs were further evaluated using fixation index (FST) and principal component analysis (PCA). Based on differences in geographical location, farming, and animal husbandry production areas/modes, with Hexi camels as the control population and Nanjiang, Sunite, Alashan, Dongjiang, Beijiang, and Qinghai camels as the selection populations for selection signature analysis. Gene Ontology (GO) classifications and Kyoto Encyclopedia of Genes and Genomes (KEGG) enrichment analyses were performed, and the functions of these genes were further studied by Genecards to identify the genes that may be related to the unique characteristics of the camel population, such as heat resistance and stress resistance. 

3.Have the authors made all data underlying the findings in their manuscript fully available?

Response: Thank you very much for your review.

In total, 1,568,087 SNPs were genotyped. FST and PCA showed that Hexi camels were genetically distant from domestic Bactrian camels in other populations. Based on differences in geographical location, farming, and animal husbandry production areas/modes, with Hexi camels as the control population and Nanjiang, Sunite, Alashan, Dongjiang, Beijiang, and Qinghai camels as the selection populations for selection signature analysis, 54, 29, 42, 34, 32, and 21 selected genes were obtained, respectively. We also found that Nanjiang (HSPA4L and INTU), Sunite (SNAI1), Alashan (INO80), and Dongjiang camels (UBE2A) harbored genes related to the environment and characteristics. 

4.Is the manuscript presented in an intelligible fashion and written in standard English?

Response: Thank you very much for your review.

Has been revised.

I have copyedited my manuscript thoroughly for language usage, spelling, and grammar.

Review Comments to the Author:

Reviewer #1:

1.In this manuscript, RAD-seq was used to detect SNP markers from seven distinct domestic Bactrian camel populations from China. SNPs were used for Fixation index (FST) and principal component (PCA) analysis, and involved genes were tested for possible selection, where their function was further studied. In my opinion, this manuscript is of importance to the scientific community and has great potential. Nevertheless, I have several suggestions and comments in which I believe it might improve the impact of it.

Response: Thank you very much for your review. I have read all the suggestions and comments carefully and answered each question one by one. 

General comments and suggestions:

1. In general, I feel that connection between sentences is missing throughout the whole text in order to create a better flow while reading. See, for example, l.50-55. I would suggest to review the whole manuscript and create better links between sentences as well as sentence construction. The discussion is also very descriptive and lacks connection between sentences (more comments about the discussion below).

Response: Thank you very much for your review.

Has been revised.

P3L46-53 and P14L288-P15L318.

These camels are also the source of various livestock products. For example, camel hair is a good textile material containing natural protein fiber [2]. Additionally, camel milk is rich in nutrition and has hypoglycemic and anticancer activities [3, 4]. Camel meat exhibits high moisture, high protein, low fat, and low cholesterol and has been shown to have medicinal and health-protective characteristics, suggesting applications in the treatment of some diseases [5-7]. 

Wensu County is close to the Taklimakan Desert. Nanjiang camels have been living in a natural environment with abundant light energy, dry climate with little rain, strong winds, and sand for a long time [32]. Thus, these camels show extremely strong heat resistance [33]. HSPA4L (encoding heat shock protein family A member 4 like) is a candidate gene related to heat resistance [29] and cellular response to heat stress. Thus, HSPA4L may be an interesting target for studying the heat resistance of Nanjiang camels. In addition, Nanjiang camels have 3–5-cm-long eyelashes; the length of the eyelashes is determined by the duration of its hair follicle growth period [34]. Inturned planar cell polarity protein (encoded by INTU) is involved in hair follicle morphogenesis [35]. Therefore, studies of INTU may provide insights into the distinct eyelash length of Nanjiang camels. Notably, Sunite camels have a thick and dense villus layer and many protective hairs, resulting in higher villus yields [36]. The key to increasing the villus yield is to ensure the full development of hair follicles [37]. SNAI1 (encoding Snail family transcriptional repressor 1) is involved in hair follicle morphogenesis [38], suggesting that SNAI1 may be an appropriate target for studying the villus yields of Sunite camels. The central production area of Alashan camels is located in the Alashan Plateau in the west of Inner Mongolia. The Alashan Plateau has strong ultraviolet rays, and Alashan camels exhibit anti-ultraviolet physiological functions, developed through a long-term evolution process [39]. INO80 (encoding INO80 complex ATPase subunit) is involved in cellular response to UV [40]. INO80 is therefore an interesting target for future experimental research. Mulei County is located in a temperate desert with strong ultraviolet rays and harsh natural ecological conditions. Dongjiang camels have also evolved physiological functions that can protect against ultraviolet radiation [41]. Because UBE2A (encoding ubiquitin-conjugating enzyme E2A) is involved in cellular response to UV [42], this gene may have applications in future studies of anti-ultraviolet radiation functions in these camel populations.

2.Some abbreviations for text comprehension have no explanation assuming all readers know what is relates to, for example l.59: MSY. I would suggest to review all abbreviations.

Response: Thank you very much for your review. I have explained and reviewed all abbreviations.

P3L62-65. 

Felkel et al. [8] generated a 3.8-Mbp assembly of the male-specific part of the Y chromosome (MSY) of Bactrian camels based on short-read next-generation sequencing data. 

3.The authors wrote several times in the different manuscript sections that FST and PCA were used for testing “phylogenetic” distances from domestic Bactrian camels in different populations. But the authors did not do any phylogenetic analysis, they have just tested for population differentiation (Fst) and individual-to-individual (PCA) genetic distances. I suggest the authors change it to “genetically” instead.

Response: Thank you very much for your review. I had changed “phylogenetic” to “genetically” instead.

4. At the beginning the taxonomic description of the domestic Bactrian camel is missing. Only in the line 59 it is stated the scientific name, but with no connection to the common name.

Response: Thank you very much for your review.

Has been revised.

P3L62-65. 

Felkel et al. [8] generated a 3.8-Mbp assembly of the male-specific part of the Y chromosome (MSY) of Bactrian camels based on short-read next-generation sequencing data. 

5. I don’t understand why the authors justify the use of Hexi camel population as “control population” for selection tests just because “these camels are genetically more distant to the other populations”, to then infer environmental pressures or production traits. I am not saying it is a bad choice, I am just commenting and calling attention to the fact that there is context information missing to understand the thought behind it. Why did the authors choose these populations specifically? What do they have in common and what do they have that differentiates them? Do all populations except for Hexi live under the same environmental conditions? Are the animals under different selective pressures on production traits on the control population compared to the others? It is always very interesting to assess genes under selection, but it is also very important to contextualize the results with the real situation, in this case, contextualize the results with the different camel populations the authors are using. I comment more on this in the “introduction” and “discussion” detailed suggestions where I suggest less descriptive text on the discussion and more information on these populations (maybe this information should start in the introduction section) to better understand the whole manuscript message.

Response: Thank you very much for your review.

Has been revised.

P1L20-P2L25 and P3L54-62.

Based on differences in geographical location, farming, and animal husbandry production areas/modes, with Hexi camels as the control population and Nanjiang, Sunite, Alashan, Dongjiang, Beijiang, and Qinghai camels as the selection populations for selection signature analysis, 54, 29, 42, 34, 32, and 21 selected genes were obtained, respectively. 

Seven domestic Bactrian camel populations are located in different production areas. The production area of Hexi camels is dominated by the farming culture of the Han nationality, whereas the production area of other domestic Bactrian camels is dominated by minority nomadic cultures. In addition, Hexi camels are located in the northern foot of the Qilian Mountains, close to the Badain Jaran Desert, and are separated from other domestic Bactrian camel populations based on geographical location. Therefore, Hexi camels can be used as a control group for exploration of the genomic resources of this species.

Detailed comments and suggestions:

Abstract.

6. L15-17: “In this study, restriction site associated DNA sequencing (RAD-seq) was used to detect SNP markers of the seven domestic Bactrian camel populations.” There are not only seven Bactrian populations in China, so I would suggest to delete “the”. Also, it is “Restriction site-associated DNA sequencing” (hyphen missing).

Response: Thank you very much for your review.

Has been revised.

P1L13-16. 

Here, we used restriction site-associated DNA sequencing (RAD-seq) to detect single nucleotide polymorphism (SNP) markers in seven domestic Bactrian camel populations. 

7.L17-20: “The filtered SNPs were used for Fixation index (FST) analysis and principal component analysis (PCA) to find out the population which is phylogenetically distant from domestic Bactrian camels in other populations as the control population.” I suggest to rephrase this sentence.

Response: Thank you very much for your review.

Has been revised.

P1L16-20. 

The identified SNPs were further evaluated using fixation index (FST) and principal component analysis (PCA). FST and PCA showed that Hexi camels were genetically distant from domestic Bactrian camels in other populations. 

Introduction.

8. L55-57: “In addition, the domestic Bactrian camel lives in hot and arid desert environment, which has potential ecological adaptation value.” Domestic Bactrian camels do not live exclusively in hot and arid, but also in cold and high altitudes. I would suggest to rephrase this sentence or to justify where specifically they live in these conditions – again, it would help if there would be already here more information on the different populations that were chosen in this study.

Response: Thank you very much for your review.

Has been revised.

P3L54-58. 

Seven domestic Bactrian camel populations are located in different production areas. The production area of Hexi camels is dominated by the farming culture of the Han nationality, whereas the production area of other domestic Bactrian camels is dominated by minority nomadic cultures. 

9. L. 58: delete “(2019)”.

Response: Thank you very much for your review.

Has been deleted.

10. L. 59: “Camelus bactrianus” should be in italic.

Response: Thank you very much for your review.

Has been revised.

P3L62-65. 

Felkel et al. [8] generated a 3.8-Mbp assembly of the male-specific part of the Y chromosome (MSY) of Bactrian camels based on short-read next-generation sequencing data. 

11.L58-66: The authors start by telling about this other study (in the 3rd person), and then on the l.63 they state “in the previous study, we used…”. I would suggest to be coherent and either write it in the 1st or 3rd person.

Response: Thank you very much for your review.

Has been revised.

P4L68-72.

Additionally, in a previous study, Liu et al. [9] used SNPs to study the genetic diversity and genetic structure of seven domestic Bactrian camel populations; in this study, the phylogenetic relationships of these populations were clarified, laying a foundation for protecting their biodiversity. 

12. L70-71: “RAD-seq analysis is becoming more and more perfect” I suggest to rephrase “more and more perfect”.

Response: Thank you very much for your review.

Has been revised.

P4L75-77. 

With the establishment of high-throughput sequencing technology and bioinformatics technology, RAD-seq analysis has become more refined and has been applied to many organisms [10].

13. L71-78: It is good that the authors state examples of studies that used the same technique in different species like eggplant or chicken, but I would suggest to state examples closer to camels or even within camelid species, that have used the same (RADseq) technique.

Response: Thank you very much for your review.

Has been revised.

P4L78-82. 

For example, Li et al. [11] used RAD-seq for genome SNP typing of 618 sows to achieve accurate prediction of genome breeding value and evaluated the accuracy of breeding value prediction. Moreover, Wang et al. [12] successfully employed RAD-seq to explore genome-wide SNPs among six breeds of Sichuan cattle. 

14. L82-87: Please add citations or links to “GO”, “KEGG” and “Genecards”.

Response: Thank you very much for your review.

Has been revised.

P4L83-L87. 

In this study, we explored genome-wide SNP markers of seven domestic Bactrian camel populations by RAD-seq, using Hexi camels as the control group for selection signature analysis. We aimed to identify genes that may be related to the unique characteristics of these camels, such as heat resistance and stress resistance. 

15. Missing a more complete sentence in the end of the introduction with the aim and importance of this study as right now these are not very clearly stated.

Response: Thank you very much for your review.

Has been revised.

P4L87-P5L90. 

Our results established interesting targets for subsequent genomics research of the domestic Bactrian camel and may facilitate genetic association analysis of economically important traits of this species.

Materials and Methods.

16. L. 123-128: substitute “readings” to “reads”.

Response: Thank you very much for your review.

Has been revised.

P6L126-P7L131.

Raw reads were processed to obtain high-quality clean reads using three strict filtering standards, as follows: (1) removing reads with greater than or equal to 10% unidentified nucleotides (N), (2) removing reads with greater than 50% of bases having phred quality scores of less than or equal to 20, and (3) removing reads aligned to the barcode adapter.

17. The authors did SNP filtering, but did the authors do sample filtering (e.g. filtering samples out with missing call of a specific rate)?

Response: Thank you very much for your review.

We didn't find abnormal samples in the process of analysis, and the variation information of each sample was generally consistent, so we didn't filter the samples.

sample snp indel

ALS-1 400909 59293

ALS-2 431342 65091

ALS-3 416516 62291

ALS-4 395045 59031

ALS-5 378303 56586

ALS-6 376948 55913

ALS-7 238629 33650

BJ-1 351434 52414

BJ-2 347812 51443

BJ-3 333301 48864

BJ-4 348118 51857

BJ-5 302660 44435

BJ-6 291205 42055

BJ-7 212065 30470

DJ-1 293884 43115

DJ-2 412285 61374

DJ-3 274259 39590

DJ-4 384356 56215

DJ-5 510888 76466

DJ-6 423925 62800

DJ-7 412384 61186

HX-1 450250 68562

HX-2 434519 65152

HX-3 405875 60727

HX-4 433282 64685

HX-5 358705 52764

NJ-1 298725 42553

NJ-2 339008 49262

NJ-3 262918 37763

NJ-4 346445 50596

NJ-5 395494 58363

NJ-6 447244 67461

NJ-7 283301 40999

QH-1 454624 68257

QH-2 410231 60948

QH-3 466470 70295

QH-4 387525 57182

QH-5 403445 58735

QH-6 379883 56187

QH-7 403622 59955

SNT-1 430952 64525

SNT-2 353668 52768

SNT-3 342415 50452

SNT-4 331900 47946

SNT-5 437473 65791

SNT-6 390642 58472

SNT-7 312674 45327

18. L. 154: are the “internal PERL scripts” available to the scientific community, to promote transparency?

Response: Thank you very much for your review.

The “internal PERL scripts” is available in Supporting information.

S1 File. The in-house PERL scripts. (DOCX)

19. I think it would be interesting to see ADMIXTURE analysis in order to infer population structure and for calculating ancestry estimates.

Response: Thank you very much for your review.

ADMIXTURE analysis in order to infer population structure and for calculating ancestry estimates is available in the literature.

Liu CM, Chen HL, Ren ZJ, Zhang CD, Yang XJ. Population genetic analysis of the domestic Bactrian camel in China by RAD-seq. Ecology and evolution. 2019;34(4): 11232-11242. doi: 10.1002/ece3.5624.

Results.

20.L. 179: I suggest to stated with how many SNP the analysis were performed.

Response: Thank you very much for your review.

Has been revised.

P9L184-186.

In total, 1,568,087 SNPs were identified in this study. After removing sites with a missing rate of 50% or more, the remaining 865,774 loci were used for FST and PCA. 

21.L. 186: The respective initials of the populations to the populations in the table description are not stated.

Response: Thank you very much for your review.

Has been revised.

P10L194-196.

BJ, Beijiang camels; DJ, Dongjiang camels; HX, Hexi camels; NJ, Nanjiang camels; QH, Qinghai camels; SNT, Sunite camels; ALS, Alashan camels.

22. L. 191: please make reference to the figure number.

Response: Thank you very much for your review.

Has been revised.

P10L203-206.

Based on the last two PCs (PC2 = 3.97%, PC3 = 3.71%), the Hexi, Nanjiang, and Alashan camels clustered together separately and were genetically distant from the domestic Bactrian camels in other populations (Fig 2).

Discussion.

23. L. 264: I would suggest to change “In the study” to “In this study”.

Response: Thank you very much for your review.

Has been revised.

P13L268-269.

In this study, 1,568,087 SNP markers from seven domestic Bactrian camel populations were detected by RAD-seq. 

24.L. 266: Fst in italic.

Response: Thank you very much for your review.

Has been revised.

P13L269-271.

The filtered SNPs were used for FST and PCA to determine the population that was genetically distant from the domestic Bactrian camel in other populations. 

25.I really feel it is missing connection between the results and the camel populations history. I think the manuscript would improve greatly if the authors would change from a descriptive discussion to a more “story telling” format around the subject. This means that maybe there should be a bit more contextualization of these populations first (starting already in the introduction), and then relate to their findings, instead of just being very descriptive in major results. This would be very helpful to better understand their findings. This also connects to one of my comments on creating better flow between sentences.

Response: Thank you very much for your review.

Has been revised.

P3L54-62 and P14L288-P15L318.

Seven domestic Bactrian camel populations are located in different production areas. The production area of Hexi camels is dominated by the farming culture of the Han nationality, whereas the production area of other domestic Bactrian camels is dominated by minority nomadic cultures. In addition, Hexi camels are located in the northern foot of the Qilian Mountains, close to the Badain Jaran Desert, and are separated from other domestic Bactrian camel populations based on geographical location. Therefore, Hexi camels can be used as a control group for exploration of the genomic resources of this species. 

Wensu County is close to the Taklimakan Desert. Nanjiang camels have been living in a natural environment with abundant light energy, dry climate with little rain, strong winds, and sand for a long time [32]. Thus, these camels show extremely strong heat resistance [33]. HSPA4L (encoding heat shock protein family A member 4 like) is a candidate gene related to heat resistance [29] and cellular response to heat stress. Thus, HSPA4L may be an interesting target for studying the heat resistance of Nanjiang camels. In addition, Nanjiang camels have 3–5-cm-long eyelashes; the length of the eyelashes is determined by the duration of its hair follicle growth period [34]. Inturned planar cell polarity protein (encoded by INTU) is involved in hair follicle morphogenesis [35]. Therefore, studies of INTU may provide insights into the distinct eyelash length of Nanjiang camels. Notably, Sunite camels have a thick and dense villus layer and many protective hairs, resulting in higher villus yields [36]. The key to increasing the villus yield is to ensure the full development of hair follicles [37]. SNAI1 (encoding Snail family transcriptional repressor 1) is involved in hair follicle morphogenesis [38], suggesting that SNAI1 may be an appropriate target for studying the villus yields of Sunite camels. The central production area of Alashan camels is located in the Alashan Plateau in the west of Inner Mongolia. The Alashan Plateau has strong ultraviolet rays, and Alashan camels exhibit anti-ultraviolet physiological functions, developed through a long-term evolution process [39]. INO80 (encoding INO80 complex ATPase subunit) is involved in cellular response to UV [40]. INO80 is therefore an interesting target for future experimental research. Mulei County is located in a temperate desert with strong ultraviolet rays and harsh natural ecological conditions. Dongjiang camels have also evolved physiological functions that can protect against ultraviolet radiation [41]. Because UBE2A (encoding ubiquitin-conjugating enzyme E2A) is involved in cellular response to UV [42], this gene may have applications in future studies of anti-ultraviolet radiation functions in these camel populations.

26. L.284-285: Please elaborate on how this finding fills in the knowledge gap.

Response: Thank you very much for your review.

Has been revised.

P14L284-287.

Overall, our findings established interesting targets for genetic association analysis of important traits of the domestic Bactrian camel in China and provided a basis for genome studies in this species [29–31].

Reviewer #2: 

Manuscript PONE-D-20-20388 seemed to be designed, based on the same RAD-seq data reported before by the same group of authors (Liu CM, Chen HL, Ren ZJ, Zhang CD, Yang XJ. Population genetic analysis of the domestic Bactrian camel in China by RAD-seq. Ecology and evolution. 2019;34(4): 11232-11242. doi: 10.1002/ece3.5624.), to examine the population differentiation and then selection signals of the seven Chinese Bactrian camels. After reading their early article published in 2019, it is hard to see the value of publishing this manuscript with limited new result.

Response: Thank you very much for your review.

This study and early article published in 2019 reflected different analysis content and purpose. The early article studied the genetic diversity and genetic structure of the domestic Bactrian camel, and conducted population genetic analysis of the species. In this study, based on differences in geographical location, farming, and animal husbandry production areas/modes, with Hexi camels as the control population and Nanjiang, Sunite, Alashan, Dongjiang, Beijiang, and Qinghai camels as the selection populations for selection signature analysis. We found that Nanjiang (HSPA4L and INTU), Sunite (SNAI1), Alashan (INO80), and Dongjiang camels (UBE2A) harbored genes related to the environment and characteristics. Our results established interesting targets for subsequent genomics research of the domestic Bactrian camel and may facilitate genetic association analysis of economically important traits of this species.

1.The manuscript is written in very poor English.

Response: Thank you very much for your review.

Has been revised.

I have copyedited my manuscript thoroughly for language usage, spelling, and grammar.

2. The data and major analyses including FST and principle component analyses were already included in the article published by the same authors.

Response: Thank you very much for your review.

In the early article, we calculated the average FST of each population. In this study, we presented the FST between the two populations. 

In this study, we identified the population which was far away from other domestic Bactrian camels in genetically based on different PCs (PC1=4.13%, PC2=3.97%, PC3=3.71%).

3. The selection signal analyses based on the π value and FST values from the comparison of the Hexi camel population with the other six camel populations make almost no sense. Considering the very low number of samples and also genetic admixture of the Hexi camels with Dongjiang, Qinghai or Sunite camels as shown in the article of 2019 and also the genetic differentiation of the Qinghai camels at PC1 in this manuscript, it is not justified to treat Hexi camels as a control population. Therefore, the annotations of the what so called functional genes are not reliable at all.

Response: Thank you very much for your review.

Seven domestic Bactrian camel populations are located in different production areas. The production area of Hexi camels is dominated by the farming culture of the Han nationality, whereas the production area of other domestic Bactrian camels is dominated by minority nomadic cultures. In addition, Hexi camels are located in the northern foot of the Qilian Mountains, close to the Badain Jaran Desert, and are separated from other domestic Bactrian camel populations based on geographical location. Therefore, Hexi camels can be used as a control group for exploration of the genomic resources of this species.

---

## [Decision Letter · Decision Letter 1]

22 Sep 2020

PONE-D-20-20388R1

Exploring the genomic resources of seven domestic Bactrian camel populations in China by restriction site-associated DNA sequencing

PLOS ONE

Dear Dr. Ren,

Thank you for submitting your manuscript to PLOS ONE. After careful consideration, we feel that it has merit but does not fully meet PLOS ONE’s publication criteria as it currently stands. Therefore, we invite you to submit a revised version of the manuscript that addresses the points raised during the review process.

We look forward to receiving your revised manuscript.

Kind regards,

Tzen-Yuh Chiang

Academic Editor

PLOS ONE

Reviewers' comments:

Reviewer's Responses to Questions

**Comments to the Author**

1. If the authors have adequately addressed your comments raised in a previous round of review and you feel that this manuscript is now acceptable for publication, you may indicate that here to bypass the “Comments to the Author” section, enter your conflict of interest statement in the “Confidential to Editor” section, and submit your "Accept" recommendation.

Reviewer #1: (No Response)

2. Is the manuscript technically sound, and do the data support the conclusions?

Reviewer #1: Partly

3. Has the statistical analysis been performed appropriately and rigorously? 

Reviewer #1: N/A

4. Have the authors made all data underlying the findings in their manuscript fully available?

Reviewer #1: Yes

5. Is the manuscript presented in an intelligible fashion and written in standard English?

Reviewer #1: No

6. Review Comments to the Author

Reviewer #1: In general, the authors have tried to address my suggestions, but not all of them. I can see an improvement on sentence connection, but I am not sure if this was enough to improve the manuscript in general. I feel the authors could have taken more time to analyse and answer to each of my suggestions, and either apply them or justify their choices. Also, I still cannot understand - and it was also mentioned by me on the first review - why the authors have chosen the Hexi population as a control population. Unfortunately, in a meaningful biological way, the justifications are not enough for a proper scientific understanding, and this part being their main finding (selection), it has to be improved.

7. PLOS authors have the option to publish the peer review history of their article (what does this mean?). If published, this will include your full peer review and any attached files.

Reviewer #1: No

---

## [Author Response · Author response to Decision Letter 1]

14 Oct 2020

PONE-D-20-20388R1

Exploring the genomic resources of seven domestic Bactrian camel populations in China by restriction site-associated DNA sequencing

PLOS ONE

Dear Editor and Reviewers,

Thank you very much for the constructive comments made by the editor and reviewers to our manuscript “Exploring the genomic resources of seven domestic Bactrian camel populations in China by restriction site-associated DNA sequencing”. I have read all the comments carefully and answered each question one by one. Where appropriate we have also included the altered or additional text after this, shown in green font, as well as line numbers in the revised manuscript. We hope this is now suitable for publication in PLoS One. Once again, thank the editor and reviewers for their constructive comments. The main comments and our specific responses are detailed below.

Sincerely,

Chenmiao Liu

(On behalf of the co-authors)

Comments to the Author

1. If the authors have adequately addressed your comments raised in a previous round of review and you feel that this manuscript is now acceptable for publication, you may indicate that here to bypass the “Comments to the Author” section, enter your conflict of interest statement in the “Confidential to Editor” section, and submit your "Accept" recommendation.

Response: Thank you very much for your review.

I have read all the comments carefully and answered each question one by one. I hope the manuscript is suitable for publication on PLoS One. 

2. Is the manuscript technically sound, and do the data support the conclusions?

Response: Thank you very much for your review.

I'm sorry about that.

P1l20-P2l25 and P3l54-l62.

① Based on differences in farming and animal husbandry production areas/modes, the production area of Hexi camels is dominated by the farming culture of the Han nationality, whereas the production area of other domestic Bactrian camels is dominated by minority nomadic cultures. Hexi camels and other domestic Bactrian camels have different production areas and modes, resulting in the selection direction differences between Hexi camels and other domestic Bactrian camels.

② According to differences in geographical location, Hexi camels are located in the northern foot of the Qilian Mountains, close to the Badain Jaran Desert, and are separated from other domestic Bactrian camel populations based on geographical location.

③ In the original project design, we used the dromedary (Camelus dromedarius) as the outgroup. However, it is difficult to find dromedary in China at present. In the next step of research, we will actively seek cooperation with foreign countries to collect camel blood samples along the Silk Road to make up for the deficiencies of current research.

3. Has the statistical analysis been performed appropriately and rigorously?

Response: Thank you very much for your review.

4. Have the authors made all data underlying the findings in their manuscript fully available?

Response: Thank you very much for your review.

5. Is the manuscript presented in an intelligible fashion and written in standard English?

Response: Thank you very much for your review.

We have used Editage professional scientific editing service to make the changes. I hope the manuscript meets the requirements of PLoS One.

Review Comments to the Author

Reviewer #1: In general, the authors have tried to address my suggestions, but not all of them. I can see an improvement on sentence connection, but I am not sure if this was enough to improve the manuscript in general. I feel the authors could have taken more time to analyse and answer to each of my suggestions, and either apply them or justify their choices. Also, I still cannot understand - and it was also mentioned by me on the first review - why the authors have chosen the Hexi population as a control population. Unfortunately, in a meaningful biological way, the justifications are not enough for a proper scientific understanding, and this part being their main finding (selection), it has to be improved.

Response: Thank you very much for your review.

(1) We have used Editage professional scientific editing service to make the changes. I hope the manuscript meets the requirements of PLoS One.

(2) I'm sorry about that.

P1l20-P2l25 and P3l54-l62.

① Based on differences in farming and animal husbandry production areas/modes, the production area of Hexi camels is dominated by the farming culture of the Han nationality, whereas the production area of other domestic Bactrian camels is dominated by minority nomadic cultures. Hexi camels and other domestic Bactrian camels have different production areas and modes, resulting in the selection direction differences between Hexi camels and other domestic Bactrian camels.

② According to differences in geographical location, Hexi camels are located in the northern foot of the Qilian Mountains, close to the Badain Jaran Desert, and are separated from other domestic Bactrian camel populations based on geographical location.

③ In the original project design, we used the dromedary (Camelus dromedarius) as the outgroup. However, it is difficult to find dromedary in China at present. In the next step of research, we will actively seek cooperation with foreign countries to collect camel blood samples along the Silk Road to make up for the deficiencies of current research.

---

## [Decision Letter · Decision Letter 2]

11 Nov 2020

PONE-D-20-20388R2

Exploring the genomic resources of seven domestic Bactrian camel populations in China by restriction site-associated DNA sequencing

PLOS ONE

Dear Dr. Ren,

Thank you for submitting your manuscript to PLOS ONE. After careful consideration, we feel that it has merit but does not fully meet PLOS ONE’s publication criteria as it currently stands. Therefore, we invite you to submit a revised version of the manuscript that addresses the points raised during the review process.

We look forward to receiving your revised manuscript.

Kind regards,

Tzen-Yuh Chiang

Academic Editor

PLOS ONE

Reviewers' comments:

Reviewer's Responses to Questions

**Comments to the Author**

1. If the authors have adequately addressed your comments raised in a previous round of review and you feel that this manuscript is now acceptable for publication, you may indicate that here to bypass the “Comments to the Author” section, enter your conflict of interest statement in the “Confidential to Editor” section, and submit your "Accept" recommendation.

Reviewer #3: (No Response)

2. Is the manuscript technically sound, and do the data support the conclusions?

Reviewer #3: Partly

3. Has the statistical analysis been performed appropriately and rigorously? 

Reviewer #3: No

4. Have the authors made all data underlying the findings in their manuscript fully available?

Reviewer #3: Yes

5. Is the manuscript presented in an intelligible fashion and written in standard English?

Reviewer #3: Yes

6. Review Comments to the Author

Reviewer #3: This manuscript has already been reviewed, as this a second revised version, however, I see it for the first time. It is rather unusual that a new reviewer is selected during such a late stage of an ongoing review process.

In the attached manuscript it is possible to see only one comment of reviewer 1. Therefore it is unfortunately not possible for me to build on and add to previous comments, but I have to provide a completely new review to the best of my knowledge.

The authors provide an interesting manuscript with the aim to identify selection signals in several groups of domestic Bactrian camels from China. However, the approach the authors took is not entirely clear.

Major comments:

*) The same or at least a very similar data set has already been published in 2019 in Ecology and Evolution by the author team (Liu et al. 2019): https://onlinelibrary.wiley.com/doi/full/10.1002/ece3.5624?af=R

Figure 2 in this manuscript and Figure 6 in Liu et al. 2019 are basically the same, and Table 1 in this manuscript is very similar to Liu et al. 2019.

As it is not good scientific practice to publish data a second time, even if they come from the same author team (self plagiarism), I suggest that it should be clearly stated that the population structure has already been identified in a previous study (Liu et al. 2019) - the sentence in l57-61 is not sufficient - and the PCA should be removed from this manuscript.

*) Selection Signals: It is not clear for me why the authors use Hexi as "control group" for exploring genomic resources (l52-53 and l71-73). What is the rationale behind this? It is clear from the previous manuscript (Liu et al. 2019) that Hexi camels are genetically more distant from the other investigated Bactrian camel groups, however, what is the meaning of "control group" here? This is not a case-control study. Why is it necessary to have a control group for detecting selection signals? To the best of my knowledge, methods for detecting signals of selection (like FST outliers, Tajima's D, Codml from PAML package, etc....) do not require a control group. What do the authors want to control for?

It is not clear why the selection signals are separately assessed for those groups, which genetically cluster together. It would be better to analyze them jointly to get a stronger signal. Otherwise, the rationale of investigating selection in the groups separately should be explained. Are there any overlapping signals of selection between the groups?

Please can the authors provide a supplementary table with a full list of the top 1% genes from the FST outlier screening, which were then subjected to GO and KEGG analyses.

*) SNP calling (l162-163): Usually a missingness rate of 10% (=90% of genotypes are present in all samples) is used in population studies using SNPs, it might be acceptable to go down to 25% of missing genotypes, however 50% of missing genotypes is not acceptable according to my point of view. What about filtering for Hardy Weinberg Equilibirum (HWE) and minor allele frequencies (MAF), which are standard filtering steps for SNPs?

Minor comments:

l53-54: The description of the Y-chromosomal resources (Felkel et al. 2019) is out of context here. It shows differentiation between domestic and wild two-humped camels, while this manuscript deals with different domestic Bactrian camel breeds. Please either remove or put into context.

l82: Please provide the concentration and the provider of the anesthetic as well as the amount (ml/kg) applied to the camels.

Figure 1: The scientific content (geographic origin of the samples) is not reflected properly in figure 1. The districts or geographical regions where the samples originate from (e.g., Gansu, Xinjiang, etc.) are not presented, while the lines of longitude and latitude do not provide any relevant information.

Fig.3-6: The figures cannot be evaluated, as the letters are too small and when zooming in, the resolution is not enough to read the text.

7. PLOS authors have the option to publish the peer review history of their article (what does this mean?). If published, this will include your full peer review and any attached files.

Reviewer #3: No

---

## [Author Response · Author response to Decision Letter 2]

23 Nov 2020

PONE-D-20-20388R2

Exploring the genomic resources of seven domestic Bactrian camel populations in China by restriction site-associated DNA sequencing

PLOS ONE

Dear Editor and Reviewer,

Thank you very much for the constructive comments made by the editor and reviewer to our manuscript “Exploring the genomic resources of seven domestic Bactrian camel populations in China by restriction site-associated DNA sequencing”. I have read all the comments carefully and answered each question one by one. Where appropriate we have also included the altered or additional text after this, shown in green font, as well as line numbers in the revised manuscript. We hope this is now suitable for publication in PLoS One. Once again, thank the editor and reviewer for their constructive comments. The main comments and our specific responses are detailed below.

Sincerely,

Chenmiao Liu

(On behalf of the co-authors)

Comments to the Author

1.If the authors have adequately addressed your comments raised in a previous round of review and you feel that this manuscript is now acceptable for publication, you may indicate that here to bypass the “Comments to the Author” section, enter your conflict of interest statement in the “Confidential to Editor” section, and submit your "Accept" recommendation.

Reviewer #3: (No Response)

Response: Thank you very much for your review.

I have read all the comments carefully and answered each question one by one. I hope the manuscript is suitable for publication on PLoS One. 

2. Is the manuscript technically sound, and do the data support the conclusions?

Reviewer #3: Partly

Response: Thank you very much for your review.

Has been revised.

(1)P1l14-17, P3l45-l48, P3l48-l51, and P1l17-20.

The identified SNPs were further evaluated using fixation index (FST), it was found that Hexi camels were genetically distant from domestic Bactrian camels in other populations.

Based on differences in farming and animal husbandry production areas/modes, the production area of Hexi camels is dominated by the farming culture of the Han nationality, whereas the production area of other domestic Bactrian camels is dominated by minority nomadic cultures. Hexi camels and other domestic Bactrian camels have different production areas and modes, resulting in the selection direction differences between Hexi camels and other domestic Bactrian camels.

According to differences in geographical location, Hexi camels are located in the northern foot of the Qilian Mountains, close to the Badain Jaran Desert, and are separated from other domestic Bactrian camel populations based on geographical location.

Based on differences in geographical location, farming, and animal husbandry production areas/modes, with Hexi camels as the reference population and Nanjiang, Sunite, Alashan, Dongjiang, Beijiang, and Qinghai camels as the target populations for selection signature analysis.

3. Has the statistical analysis been performed appropriately and rigorously?

Reviewer #3: No

Response: Thank you very much for your review.

Has been revised.

We used Hexi camels as the reference population and Nanjiang, Sunite, Alashan, Dongjiang, Beijiang, and Qinghai camels as the target populations for selection signature analysis. FST and π were used to select the top 1% regions. 54, 29, 42, 34, 32, and 21 selected genes, respectively, were obtained. GO classification and KEGG enrichment analysis were carried out on the selected genes. 

(S2-S7 Table. The results of the selection signature analysis. S8-S13 Table. Go classification of the selected genes of domestic Bactrian camels. S14-S19 Table. KEGG enrichment of the selected genes of domestic Bactrian camels.)

Review Comments to the Author

Reviewer #3: This manuscript has already been reviewed, as this a second revised version, however, I see it for the first time. It is rather unusual that a new reviewer is selected during such a late stage of an ongoing review process.

In the attached manuscript it is possible to see only one comment of reviewer 1. Therefore it is unfortunately not possible for me to build on and add to previous comments, but I have to provide a completely new review to the best of my knowledge.

The authors provide an interesting manuscript with the aim to identify selection signals in several groups of domestic Bactrian camels from China. However, the approach the authors took is not entirely clear.

Response: Thank you very much for your review.

Has been revised.

P1l17-20

Based on differences in geographical location, farming, and animal husbandry production areas/modes, with Hexi camels as the reference population and Nanjiang, Sunite, Alashan, Dongjiang, Beijiang, and Qinghai camels as the target populations for selection signature analysis. FST and π were used to select the top 1% regions. 54, 29, 42, 34, 32, and 21 selected genes, respectively, were obtained. 

Major comments:

1.The same or at least a very similar data set has already been published in 2019 in Ecology and Evolution by the author team (Liu et al. 2019): https://onlinelibrary.wiley.com/doi/full/10.1002/ece3.5624?af=R Figure 2 in this manuscript and Figure 6 in Liu et al. 2019 are basically the same, and Table 1 in this manuscript is very similar to Liu et al. 2019. As it is not good scientific practice to publish data a second time, even if they come from the same author team (self plagiarism), I suggest that it should be clearly stated that the population structure has already been identified in a previous study (Liu et al. 2019) - the sentence in l57-61 is not sufficient - and the PCA should be removed from this manuscript.

Response: Thank you very much for your review.

Has been revised.

P1l14-17, P3l56-59, and P11l216-220

The identified SNPs were further evaluated using fixation index (FST), it was found that Hexi camels were genetically distant from domestic Bactrian camels in other populations. 

Additionally, Liu et al. [9] used single nucleotide polymorphisms (SNPs) to study the genetic diversity and genetic structure of seven domestic Bactrian camel populations; in this study, the phylogenetic relationships of these populations were clarified, laying a foundation for protecting their biodiversity.

In the previous PCA and population structure analysis, we have already been identified the population structure, and Hexi camels were obviously separated from other camel populations [9]. FST using filtered SNPs showed that Hexi camels were genetically distant from the domestic Bactrian camel in other populations.

2.Selection Signals: It is not clear for me why the authors use Hexi as "control group" for exploring genomic resources (l52-53 and l71-73). What is the rationale behind this? It is clear from the previous manuscript (Liu et al. 2019) that Hexi camels are genetically more distant from the other investigated Bactrian camel groups, however, what is the meaning of "control group" here? This is not a case-control study. Why is it necessary to have a control group for detecting selection signals? To the best of my knowledge, methods for detecting signals of selection (like FST outliers, Tajima's D, Codml from PAML package, etc....) do not require a control group. What do the authors want to control for?

It is not clear why the selection signals are separately assessed for those groups, which genetically cluster together. It would be better to analyze them jointly to get a stronger signal. Otherwise, the rationale of investigating selection in the groups separately should be explained. Are there any overlapping signals of selection between the groups?

Please can the authors provide a supplementary table with a full list of the top 1% genes from the FST outlier screening, which were then subjected to GO and KEGG analyses.

Response: Thank you very much for your review.

Has been revised.

(1)P3l45-l48, P3l48-l51, and P1l17-20.

① Based on differences in farming and animal husbandry production areas/modes, the production area of Hexi camels is dominated by the farming culture of the Han nationality, whereas the production area of other domestic Bactrian camels is dominated by minority nomadic cultures. Hexi camels and other domestic Bactrian camels have different production areas and modes, resulting in the selection direction differences between Hexi camels and other domestic Bactrian camels.

② According to differences in geographical location, Hexi camels are located in the northern foot of the Qilian Mountains, close to the Badain Jaran Desert, and are separated from other domestic Bactrian camel populations based on geographical location.

③ Based on differences in geographical location, farming, and animal husbandry production areas/modes, with Hexi camels as the reference population and Nanjiang, Sunite, Alashan, Dongjiang, Beijiang, and Qinghai camels as the target populations for selection signature analysis.

(2) "control group" is "reference group". There are differences between Hexi camels and the other six domestic Bactrian camel populations in geographical location, farming, and animal husbandry production areas/modes. We used Hexi camels as the reference population and the other six domestic Bactrian camel populations as the target populations for selection signature analysis.

(3) Because the characteristics of genetic diversity of different populations are different, our aim is to find out the selected range of other populations relative to Hexi camel population.

Diao SQ, Huang SW, Chen ZT, Teng JY, Ma YL, Yuan XL, et al. Genome-Wide Signatures of Selection Detection in Three South China Indigenous Pigs. Genes. 2019;10(5): 346. doi: 10.3390/genes10050346.

Yang J, Li WR, Lv FH, He SG, Tian SL, Peng WF, et al. Whole-Genome Sequencing of Native Sheep Provides Insights into Rapid Adaptations to Extreme Environments. Molecular Biology and Evolution. 2016;33(10): 2576-2592. doi: 10.1093/molbev/msw129.

(4)There are differences between Hexi camels and the other six domestic Bactrian camel populations in geographical location, farming, and animal husbandry production areas/modes. We want to find out the selected range of the other six domestic Bactrian camel populations relative to Hexi camel population, so we can judge the selected range by the nucleotide diversity within different populations, whether it deviates from neutral evolution or not, and FST between populations.

Diao SQ, Huang SW, Chen ZT, Teng JY, Ma YL, Yuan XL, et al. Genome-Wide Signatures of Selection Detection in Three South China Indigenous Pigs. Genes. 2019;10(5): 346. doi: 10.3390/genes10050346.

Yang J, Li WR, Lv FH, He SG, Tian SL, Peng WF, et al. Whole-Genome Sequencing of Native Sheep Provides Insights into Rapid Adaptations to Extreme Environments. Molecular Biology and Evolution. 2016;33(10): 2576-2592. doi: 10.1093/molbev/msw129.

(5)According to our previous studies (PCA and Structure), there are still some differences in genetic clustering among the other six domestic Bactrian camel populations. In addition, the six domestic Bactrian camel populations are located in different geographical locations and are far away from each other. 

There aren't overlapping signals of selection between the populations.

Liu CM, Chen HL, Ren ZJ, Zhang CD, Yang XJ. Population genetic analysis of the domestic Bactrian camel in China by RAD-seq. Ecology and Evolution. 2019;9(19): 11232-11242. doi: 10.1002/ece3.5624.

(6)FST and π were used to select the top 1% regions. Supplementary tables are provided in the supporting information. (S2-S7 Table. The results of the selection signature analysis. S8-S13 Table. Go classification of the selected genes of domestic Bactrian camels. S14-S19 Table. KEGG enrichment of the selected genes of domestic Bactrian camels.)

3.SNP calling (l162-163): Usually a missingness rate of 10% (=90% of genotypes are present in all samples) is used in population studies using SNPs, it might be acceptable to go down to 25% of missing genotypes, however 50% of missing genotypes is not acceptable according to my point of view. What about filtering for Hardy Weinberg Equilibirum (HWE) and minor allele frequencies (MAF), which are standard filtering steps for SNPs?

Response: Thank you very much for your review.

Has been revised.

（1）With reference to some literature, we used 50% of missing genotypes. 

Duggal P, Gillanders EM, Holmes TN, Bailey-Wilson JE. Establishing an adjusted p-value threshold to control the family-wide type 1 error in genome wide association studies. BMC genomics. 2008;9(1): 516. doi: 10.1186/1471-2164-9-516.

Marandel F, Charrier G, Lamy JB, Le Cam S, Lorance P, Trenkel VM. Estimating effective population size using RADseq: Effects of SNP selection and sample size. Ecology and Evolution. 2019;10(4): 1929-1937. doi: 10.1002/ece3.6016.

Waters CD, Hard JJ, Brieuc MS, Fast DE, Warheit KI, Waples RS, et al. Effectiveness of managed gene flow in reducing genetic divergence associated with captive breeding. Evolutionary applications. 2015;8(10): 956-971. doi: 10.1111/eva.12331.

（2）We didn't filter whether the loci conform to Hardy Weinberg Equilibirum (HWE), because "Hardy Weinberg Equilibirum" means that under ideal conditions, the frequency of each allele is stable in heredity, that is to say, the gene balance is maintained. The conditions of the law are as follows: (1) the population is large enough, (2) there is random mating between individuals in the population, (3) there is no mutation, (4) there is no selection, (5) there is no migration, and (6) there is no genetic drift. The theoretical basis of our analysis is based on the existence of factors such as selection and drift.

In this study, GATK’s Unified Genotyper was used to conduct variant calling on all samples, and GATK Variant Filtration with proper standards was used to filter SNPs (-Window 4, -filter “QD < 2.0 || FS > 60.0 || MQ <40.0”, -G_filter “GQ < 20”) [19].

Ren AY, Du K, Jia XB, Yang R, Wang J, Chen SY, et al. Genetic diversity and population structure of four Chinese rabbit breeds. PLOS ONE. 2019;14(9): e0222503. doi: 10.1371/journal.pone.0222503.

Minor comments:

1. l53-54: The description of the Y-chromosomal resources (Felkel et al. 2019) is out of context here. It shows differentiation between domestic and wild two-humped camels, while this manuscript deals with different domestic Bactrian camel breeds. Please either remove or put into context.

Response: Thank you very much for your review.

Has been revised.

Bai et al. [8] analyzed the genetic diversity of six domestic Bactrian camel populations in China by using microsatellite markers technology, and understood the genetic diversity of domestic Bactrian camel populations in China and the genetic evolutionary relationships between different populations.

2.l82: Please provide the concentration and the provider of the anesthetic as well as the amount (ml/kg) applied to the camels.

Response: Thank you very much for your review.

Has been revised.

During the sampling process, Miannaining was used, which is a compound preparation of xylazine and dihydroetorphine hydrochloride, with a specification of 2 ml per tube, and was purchased from the Institute of Quartermaster University of the Chinese People's Liberation Army. The application amount of camel was 2.5-3.5ml/100kg, and each sample was derived from a different family; there was no kinship among individuals.

3. Figure 1: The scientific content (geographic origin of the samples) is not reflected properly in figure 1. The districts or geographical regions where the samples originate from (e.g., Gansu, Xinjiang, etc.) are not presented, while the lines of longitude and latitude do not provide any relevant information.

Response: Thank you very much for your review.

Has been revised.

4. Fig.3-6: The figures cannot be evaluated, as the letters are too small and when zooming in, the resolution is not enough to read the text.

Response: Thank you very much for your review.

Has been revised.

---

## [Decision Letter · Decision Letter 3]

26 Nov 2020

PONE-D-20-20388R3

Exploring the genomic resources of seven domestic Bactrian camel populations in China by restriction site-associated DNA sequencing

PLOS ONE

Dear Dr. Ren,

Thank you for submitting your manuscript to PLOS ONE. After careful consideration, we feel that it has merit but does not fully meet PLOS ONE’s publication criteria as it currently stands. Therefore, we invite you to submit a revised version of the manuscript that addresses the points raised during the review process.

We look forward to receiving your revised manuscript.

Kind regards,

Tzen-Yuh Chiang

Academic Editor

PLOS ONE

Reviewers' comments:

Reviewer's Responses to Questions

**Comments to the Author**

1. If the authors have adequately addressed your comments raised in a previous round of review and you feel that this manuscript is now acceptable for publication, you may indicate that here to bypass the “Comments to the Author” section, enter your conflict of interest statement in the “Confidential to Editor” section, and submit your "Accept" recommendation.

Reviewer #3: (No Response)

2. Is the manuscript technically sound, and do the data support the conclusions?

Reviewer #3: Partly

3. Has the statistical analysis been performed appropriately and rigorously? 

Reviewer #3: No

4. Have the authors made all data underlying the findings in their manuscript fully available?

Reviewer #3: Yes

5. Is the manuscript presented in an intelligible fashion and written in standard English?

Reviewer #3: No

6. Review Comments to the Author

Reviewer #3: I would like to thank the authors for their efforts to revise the manuscript.

Unfortunately many of my comments were not considered in the manuscript text and the authors' explanations why they did not consider them are not satisfactorily.

I just can repeat my previous comments that the authors still include previously published data in this manuscript, which does not follow good scientific practice.

Either the authors show convincingly what is different in the data set presented or they need to remove all of the previously published data (Table 2, l165-189).

Similarly, the filtering of the SNP data set is not performed sufficiently rigorous to ensure robust results for their research question, namely selection in the different camel populations. See my comments below in "Methods".

It is clear that there is larger genetic difference between the Hexi camelsbut again, there is no rationale to use it as "control" camel group is used as a reference explained in the manuscript text, now it is only included in the answer- Hexi camels can be treated like the other camel groups, and FST outliers - to detect signals of selection can be estimated and discussed in the light of the different production traits and geographical location.

Moreover, higher FSTs between Hexi and the other camel populations can just be signal of drift and geographic isolation. How do the authors

There is no logic behind a "control" group.

Introduction:

l53-57: The Y-chromosomal SNP analysis is still out of the context here. What does the Y-chromosomal differentiation between wild and domestic camels have to do with the differentiation among domestic camels?

Methods:

I acknowledge that that there are different approaches for data filtering which depend on the research question. The paper, which is cited by the authors, e.g, doi: 10.1002/ece3.6016., has a completely different research question. In that paper actually the "potential heterozygote miscall rate was estimated by comparing the observed number of heterozygous individuals with the number expected given the allele frequency and assuming the SNP was in Hardy–Weinberg equilibrium. Based on this the 50% missing rate was inferred. Based on this a data set containing only genotypes with read depths between 30 and 300 copies, MAF ≥0.01 and NAs ≤0.5 was created." On top, the authors used another data set with "only" 25% of missing genotypes to answer their questions to "determine the effects of the MAF, the proportion of missing data and the number of SNPs on Ne estimates when applying the LD approach to RADseq data." And actually they detected a difference of Ne estimates up to 7.5-fold using different filtering thresholds: "Depending on the combination of threshold values, Ne estimates varied by up to a factor of 7.5."

Filtering SNPs which below a certain HWE value (usually 10-6 or using an FDR as cut-off is a method to account for genotyping errors. The same accounts for filtering for a minor allele frequency, which the authors completely ignore.

Here, the authors look for signatures of selection, which actually requires a rigorously filtered dataset to avoid false positive signals. If they want to include SNPs which are actually not present in 50% of the dataset, they have to justify this approach.

Therefore, I encourage the authors to repeat their analysis with a less amount of missing genotyping data and show that the same results can be retrieved with both datasets.

Results:

The authors still present already published results and simply removed Figure 2 from the figures, but not as results from the text (Table 2, l165-189), even the PCA is still mentioned as Fig.2. All results, which have been published previously need to be removed.

English correction is necessary throughout the manuscript.

7. PLOS authors have the option to publish the peer review history of their article (what does this mean?). If published, this will include your full peer review and any attached files.

Reviewer #3: No

---

## [Author Response · Author response to Decision Letter 3]

1 Mar 2021

PONE-D-20-20388R3

Exploring the genomic resources of seven domestic Bactrian camel populations in China through restriction site-associated DNA sequencing

PLOS ONE

Dear Editor and Reviewer,

Thank you very much for the constructive comments made by the editor and reviewer to our manuscript “Exploring the genomic resources of seven domestic Bactrian camel populations in China through restriction site-associated DNA sequencing”. I have read all the comments carefully and answered each question one by one. Where appropriate we have also included the altered or additional text after this, shown in green font, as well as line numbers in the revised manuscript. We hope this is now suitable for publication in PLoS One. Once again, thank the editor and reviewer for their constructive comments. The main comments and our specific responses are detailed below.

Sincerely,

Chenmiao Liu

(On behalf of the co-authors)

Comments to the Author

1.If the authors have adequately addressed your comments raised in a previous round of review and you feel that this manuscript is now acceptable for publication, you may indicate that here to bypass the “Comments to the Author” section, enter your conflict of interest statement in the “Confidential to Editor” section, and submit your "Accept" recommendation.

Reviewer #3: (No Response)

Response: Thank you very much for your review.

I have read all the comments carefully and answered each question one by one. I hope the manuscript is suitable for publication on PLoS One. 

2. Is the manuscript technically sound, and do the data support the conclusions?

Reviewer #3: Partly

Response: Thank you very much for your review.

Has been revised.

(1)P6l125-P7l128 and P10l207-208.

Finally, the obtained SNPs were filtered with VCFtools (https://github.com/vcftools/vcftools) for further analysis with Minor Allele Frequency (MAF) > 0.05 and proportion of missing genotyping data < 25% as parameters. This data file was then used in subsequent analyses [22].

In this study, we used RAD-seq to detect 482,786 SNP markers from seven domestic Bactrian camel populations.

(2)P1l14-18, P3l46-55 and P10l208-212.

Based on differences in geographical location, farming, and animal husbandry production areas/modes, the Hexi camels were selected as the reference population, and the Nanjiang, Sunite, Alashan, Dongjiang, Beijiang, and Qinghai camels were the target populations for selection signature analysis.

Seven domestic Bactrian camel populations come from different regions of China. Hexi camels come from an area dominated by Han farming culture, whereas the other domestic Bactrian camels originate from regions dominated by minority nomadic culture. Hexi camels are found in the northern foot of the Qilian Mountains, close to the Badain Jaran Desert, and are therefore geographically separated from other domestic Bactrian camel populations. In addition, Nanjiang camels have stronger heat resistance than Hexi camels, Sunite camels have higher villus yields than Hexi camels, Alashan camels and Dongjiang camels form stronger anti-ultraviolet physiological functions than Hexi camels. Thus, Hexi camels can be used as a reference group for exploration of the genomic resources of their species. 

Yongchang County, where the Hexi camels originate from, is located at the northern foot of the Qilian Mountains, close to the Badain Jaran Desert, and is dominated by Han farming culture. Based on differences in geographical location, farming, and animal husbandry production areas/modes, Hexi camels were used as the reference population for selection signature analysis.

3. Has the statistical analysis been performed appropriately and rigorously?

Reviewer #3: No

Response: Thank you very much for your review.

Has been revised.

P6l125-P7l128 and P10l207-208.

Finally, the obtained SNPs were filtered with VCFtools (https://github.com/vcftools/vcftools) for further analysis with Minor Allele Frequency (MAF) > 0.05 and proportion of missing genotyping data < 25% as parameters. This data file was then used in subsequent analyses [22].

In this study, we used RAD-seq to detect 482,786 SNP markers from seven domestic Bactrian camel populations.

4. Have the authors made all data underlying the findings in their manuscript fully available?

Reviewer #3: Yes

Response: Thank you very much for your review.

5.Is the manuscript presented in an intelligible fashion and written in standard English?

Reviewer #3: No

Response: Thank you very much for your review.

We have used Editage professional scientific editing service to make the changes. I hope the manuscript meets the requirements of PLoS One.

Review Comments to the Author

Reviewer #3: 

(1)I would like to thank the authors for their efforts to revise the manuscript. Unfortunately many of my comments were not considered in the manuscript text and the authors' explanations why they did not consider them are not satisfactorily.

Response: Thank you very much for your review.

I'm terribly sorry about that. I have read all the comments carefully and answered each question one by one.

(2)I just can repeat my previous comments that the authors still include previously published data in this manuscript, which does not follow good scientific practice. Either the authors show convincingly what is different in the data set presented or they need to remove all of the previously published data (Table 2, l165-189).

Response: Thank you very much for your review.

I'm terribly sorry about that. I have removed all of the previously published data.

(3)Similarly, the filtering of the SNP data set is not performed sufficiently rigorous to ensure robust results for their research question, namely selection in the different camel populations. See my comments below in "Methods".

Response: Thank you very much for your review.

Has been revised.

We repeated the analysis with a less amount of missing genotyping data and a minor allele frequency.

P6l125-P7l128 and P10l207-208.

Finally, the obtained SNPs were filtered with VCFtools (https://github.com/vcftools/vcftools) for further analysis with Minor Allele Frequency (MAF) > 0.05 and proportion of missing genotyping data < 25% as parameters. This data file was then used in subsequent analyses [22].

In this study, we used RAD-seq to detect 482,786 SNP markers from seven domestic Bactrian camel populations.

(4)It is clear that there is larger genetic difference between the Hexi camelsbut again, there is no rationale to use it as "control" camel group is used as a reference explained in the manuscript text, now it is only included in the answer- Hexi camels can be treated like the other camel groups, and FST outliers - to detect signals of selection can be estimated and discussed in the light of the different production traits and geographical location. Moreover, higher FSTs between Hexi and the other camel populations can just be signal of drift and geographic isolation. How do the authors There is no logic behind a "control" group.

Response: Thank you very much for your review.

Has been revised.

P1l14-18, P3l46-55 and P10l208-212.

Based on differences in geographical location, farming, and animal husbandry production areas/modes, the Hexi camels were selected as the reference population, and the Nanjiang, Sunite, Alashan, Dongjiang, Beijiang, and Qinghai camels were the target populations for selection signature analysis.

Seven domestic Bactrian camel populations come from different regions of China. Hexi camels come from an area dominated by Han farming culture, whereas the other domestic Bactrian camels originate from regions dominated by minority nomadic culture. Hexi camels are found in the northern foot of the Qilian Mountains, close to the Badain Jaran Desert, and are therefore geographically separated from other domestic Bactrian camel populations. In addition, Nanjiang camels have stronger heat resistance than Hexi camels, Sunite camels have higher villus yields than Hexi camels, Alashan camels and Dongjiang camels form stronger anti-ultraviolet physiological functions than Hexi camels. Thus, Hexi camels can be used as a reference group for exploration of the genomic resources of their species.

Yongchang County, where the Hexi camels originate from, is located at the northern foot of the Qilian Mountains, close to the Badain Jaran Desert, and is dominated by Han farming culture. Based on differences in geographical location, farming, and animal husbandry production areas/modes, Hexi camels were used as the reference population for selection signature analysis.

Introduction:

l53-57: The Y-chromosomal SNP analysis is still out of the context here. What does the Y-chromosomal differentiation between wild and domestic camels have to do with the differentiation among domestic camels?

Response: Thank you very much for your review.

Has been revised.

P3l55-59.

Bai et al. [8] analyzed the genetic diversity of six domestic Bactrian camel populations in China using microsatellite marker technology, gaining an understanding of the genetic diversity of domestic Bactrian camel populations in China, as well as the evolutionary relationships between different populations.

Bai L, Zhou JW, Li XH, Feng DZ. Analysis on the genetic diversity of six Bactrian camel populations in China using microsatellite marker. China Animal Husbandry & Veterinary Medicine. 2017;44(6): 1734-1745. doi: 10.16431/j.cnki. 1671-7236.2017.06.023.

Methods:

Filtering SNPs which below a certain HWE value (usually 10-6 or using an FDR as cut-off is a method to account for genotyping errors. The same accounts for filtering for a minor allele frequency, which the authors completely ignore.

Here, the authors look for signatures of selection, which actually requires a rigorously filtered dataset to avoid false positive signals. If they want to include SNPs which are actually not present in 50% of the dataset, they have to justify this approach. Therefore, I encourage the authors to repeat their analysis with a less amount of missing genotyping data and show that the same results can be retrieved with both datasets.

Response: Thank you very much for your review.

Has been revised.

(1)Sites that don't conform to the Hardy-Weinberg equilibrium may also be selected. 

①Ren AY, Du K, Jia XB, Yang R, Wang J, Chen SY, et al. Genetic diversity and population structure of four Chinese rabbit breeds. PLOS ONE. 2019;14(9): e0222503. doi: 10.1371/journal.pone.0222503.

②Sun T, Huang GY, Wang ZH, Teng SH, Cao YH, Sun JL, et al. Selection signatures of Fuzhong Buffalo based on whole-genome sequences. BMC Genomics. 2020;21(1): 674. doi: 10.1186/s12864-020-07095-8.

(2)We repeated the analysis with a less amount of missing genotyping data and a minor allele frequency.

P6l125-P7l128 and P10l207-208.

Finally, the obtained SNPs were filtered with VCFtools (https://github.com/vcftools/vcftools) for further analysis with Minor Allele Frequency (MAF) > 0.05 and proportion of missing genotyping data < 25% as parameters. This data file was then used in subsequent analyses [22].

In this study, we used RAD-seq to detect 482,786 SNP markers from seven domestic Bactrian camel populations.

Results:

(1)The authors still present already published results and simply removed Figure 2 from the figures, but not as results from the text (Table 2, l165-189), even the PCA is still mentioned as Fig.2. All results, which have been published previously need to be removed.

Response: Thank you very much for your review.

I'm terribly sorry about that. I have removed all of the previously published data.

(2)English correction is necessary throughout the manuscript.

Response: Thank you very much for your review.

We have used Editage professional scientific editing service to make the changes. I hope the manuscript meets the requirements of PLoS One.

---

## [Decision Letter · Decision Letter 4]

9 Mar 2021

PONE-D-20-20388R4

Exploring the genomic resources of seven domestic Bactrian camel populations in China through restriction site-associated DNA sequencing

PLOS ONE

Dear Dr. Ren,

Thank you for submitting your manuscript to PLOS ONE. After careful consideration, we feel that it has merit but does not fully meet PLOS ONE’s publication criteria as it currently stands. Therefore, we invite you to submit a revised version of the manuscript that addresses the points raised during the review process.

We look forward to receiving your revised manuscript.

Kind regards,

Tzen-Yuh Chiang

Academic Editor

PLOS ONE

Reviewers' comments:

Reviewer's Responses to Questions

**Comments to the Author**

1. If the authors have adequately addressed your comments raised in a previous round of review and you feel that this manuscript is now acceptable for publication, you may indicate that here to bypass the “Comments to the Author” section, enter your conflict of interest statement in the “Confidential to Editor” section, and submit your "Accept" recommendation.

Reviewer #4: (No Response)

2. Is the manuscript technically sound, and do the data support the conclusions?

Reviewer #4: No

3. Has the statistical analysis been performed appropriately and rigorously? 

Reviewer #4: No

4. Have the authors made all data underlying the findings in their manuscript fully available?

Reviewer #4: Yes

5. Is the manuscript presented in an intelligible fashion and written in standard English?

Reviewer #4: Yes

6. Review Comments to the Author

Reviewer #4: Major

1) I agree with the Reviewer#3 that choosing Hexi camels as the reference for selection analysis is not appropriate. Even if Hexi camels are geographically, genetically and phenotypically different from other camels, there is no evidence that the other populations are derived from Hexi camels. I think a better strategy is to compare each population with the pool of all remaining others.

2) In regard to the SNP quality, I think an independent dataset is needed to justify the authors’ filtering strategy. In fact, there are a lot of WGS data for domesticated Bactrian camels (e.g. doi: 10.1038/s42003-019-0734-6). It is surprising that the authors did not mention any recent research progress of the camel genome.

3) P3:51-54 The authors listed many phenotype differences among the populations such as heat resistance, villus yields, but there is no data or reference support the claims.

Minor

Text in figures is not friendly to read. For example, what does “-log10({\\pi_{N}})” mean?

7. PLOS authors have the option to publish the peer review history of their article (what does this mean?). If published, this will include your full peer review and any attached files.

Reviewer #4: No

---

## [Author Response · Author response to Decision Letter 4]

28 Mar 2021

PONE-D-20-20388R4

Exploring the genomic resources of seven domestic Bactrian camel populations in China through restriction site-associated DNA sequencing

PLOS ONE

Dear Editor and Reviewer,

Thank you very much for the constructive comments made by the editor and reviewer to our manuscript “Exploring the genomic resources of seven domestic Bactrian camel populations in China through restriction site-associated DNA sequencing”. I have read all the comments carefully and answered each question one by one. Where appropriate we have also included the altered or additional text after this, shown in red font, as well as line numbers in the revised manuscript. We hope this is now suitable for publication in PLoS One. Once again, thank the editor and reviewer for their constructive comments. The main comments and our specific responses are detailed below.

Sincerely,

Chenmiao Liu

(On behalf of the co-authors)

Comments to the Author

1. If the authors have adequately addressed your comments raised in a previous round of review and you feel that this manuscript is now acceptable for publication, you may indicate that here to bypass the “Comments to the Author” section, enter your conflict of interest statement in the “Confidential to Editor” section, and submit your "Accept" recommendation.

Reviewer #4: (No Response)

Response: Thank you very much for your review.

I have read all the comments carefully and answered each question one by one. I hope the manuscript is suitable for publication on PLoS One. 

2. Is the manuscript technically sound, and do the data support the conclusions?

Reviewer #4: No

Response: Thank you very much for your review.

Has been revised.

P1l14-18 and P3l49-52.

The pool of all remaining others were selected as the reference population, and the Nanjiang, Sunite, Alashan, Dongjiang, Beijiang, Qinghai, and Hexi camels were the target populations for selection signature analysis. We obtained 603, 494, 622, 624, 444, 588, and 762 selected genes, respectively, from members of the seven target populations.

Nanjiang camels have stronger heat resistance than the pool of all remaining others, Alashan camels form stronger anti-ultraviolet physiological functions than the pool of all remaining others. Thus, the pool of all remaining others can be used as a reference group for exploration of the genomic resources of their species. 

3. Has the statistical analysis been performed appropriately and rigorously?

Reviewer #4: No

Response: Thank you very much for your review.

P1l14-18, P3l49-52 and P8l154-157.

The pool of all remaining others were selected as the reference population, and the Nanjiang, Sunite, Alashan, Dongjiang, Beijiang, Qinghai, and Hexi camels were the target populations for selection signature analysis. We obtained 603, 494, 622, 624, 444, 588, and 762 selected genes, respectively, from members of the seven target populations.

Nanjiang camels have stronger heat resistance than the pool of all remaining others, Alashan camels form stronger anti-ultraviolet physiological functions than the pool of all remaining others. Thus, the pool of all remaining others can be used as a reference group for exploration of the genomic resources of their species.

Using the pool of all remaining others as the reference population and Nanjiang, Sunite, Alashan, Dongjiang, Beijiang, Qinghai, and Hexi camels as the target populations, 603, 494, 622, 624, 444, 588, and 762 selected genes, respectively, were obtained.

4. Have the authors made all data underlying the findings in their manuscript fully available?

Reviewer #4: Yes

Response: Thank you very much for your review.

5. Is the manuscript presented in an intelligible fashion and written in standard English?

Reviewer #4: Yes

Response: Thank you very much for your review.

Reviewer #4: Major

1) I agree with the Reviewer#3 that choosing Hexi camels as the reference for selection analysis is not appropriate. Even if Hexi camels are geographically, genetically and phenotypically different from other camels, there is no evidence that the other populations are derived from Hexi camels. I think a better strategy is to compare each population with the pool of all remaining others.

Response: Thank you very much for your review.

Has been revised. We chose a better strategy is to compare each population with the pool of all remaining others.

P1l14-18 and P3l49-52.

The pool of all remaining others were selected as the reference population, and the Nanjiang, Sunite, Alashan, Dongjiang, Beijiang, Qinghai, and Hexi camels were the target populations for selection signature analysis. We obtained 603, 494, 622, 624, 444, 588, and 762 selected genes, respectively, from members of the seven target populations.

Nanjiang camels have stronger heat resistance than the pool of all remaining others, Alashan camels form stronger anti-ultraviolet physiological functions than the pool of all remaining others. Thus, the pool of all remaining others can be used as a reference group for exploration of the genomic resources of their species. 

2) In regard to the SNP quality, I think an independent dataset is needed to justify the authors’ filtering strategy. In fact, there are a lot of WGS data for domesticated Bactrian camels (e.g. doi: 10.1038/s42003-019-0734-6). It is surprising that the authors did not mention any recent research progress of the camel genome.

Response: Thank you very much for your review.

Has been revised. 

(1)SNPs were re-filtered in accordance with the requirements of the last reviewer. In addition, we also refered to the related literature published on PLoS One.

Basak M, Uzun B, Yol E. Genetic diversity and population structure of the Mediterranean sesame core collection with use of genome-wide SNPs developed by double digest RAD-Seq. PLoS One. 2019;14(10): e0223757. doi: 10.1371/journal.pone.0223757.

Ren AY, Du K, Jia XB, Yang R, Wang J, Chen SY, et al. Genetic diversity and population structure of four Chinese rabbit breeds. PLoS One. 2019;14(9): e0222503. doi: 10.1371/journal.pone.0222503.

(2)At present, some studies have been conducted on domestic Bactrian camels. Ming et al. [8] sequenced the whole genome of 128 camels across Asia and revealed origin and migration of domestic Bactrian camels.

3) P3:51-54 The authors listed many phenotype differences among the populations such as heat resistance, villus yields, but there is no data or reference support the claims.

Response: Thank you very much for your review.

Has been revised. 

Nanjiang camels have stronger heat resistance than the pool of all remaining others, Alashan camels form stronger anti-ultraviolet physiological functions than the pool of all remaining others[8, 9].

8. Tian YZ, Nurbia U, Wang LJ, Wu WW, Xu XM, Zhang YH, et al. Genetic diversity analysis of Bactrian camel in six places of Xinjiang. Animal Husbandry & Veterinary Medicine. 2012;44(6): 38-43.

9. Wang JM, Cui PJ, Zhong YM, Li JW, Chu JM. Species richness pattern and its environmental interpretation in the plant region of Alashan Plateau. Journal of Beijing Forestry University. 2019;41(3): 14-23.

Minor

Text in figures is not friendly to read. For example, what does “-log10({\\pi_{N}})” mean?

Response: Thank you very much for your review.

Text in figures has been revised.

---

## [Decision Letter · Decision Letter 5]

1 Apr 2021

Exploring the genomic resources of seven domestic Bactrian camel populations in China through restriction site-associated DNA sequencing

PONE-D-20-20388R5

Dear Dr. Ren,

We’re pleased to inform you that your manuscript has been judged scientifically suitable for publication and will be formally accepted for publication once it meets all outstanding technical requirements.

Kind regards,

Tzen-Yuh Chiang

Academic Editor

PLOS ONE

Additional Editor Comments (optional):

Reviewers' comments:

Reviewer's Responses to Questions

**Comments to the Author**

1. If the authors have adequately addressed your comments raised in a previous round of review and you feel that this manuscript is now acceptable for publication, you may indicate that here to bypass the “Comments to the Author” section, enter your conflict of interest statement in the “Confidential to Editor” section, and submit your "Accept" recommendation.

Reviewer #4: (No Response)

2. Is the manuscript technically sound, and do the data support the conclusions?

Reviewer #4: (No Response)

3. Has the statistical analysis been performed appropriately and rigorously? 

Reviewer #4: (No Response)

4. Have the authors made all data underlying the findings in their manuscript fully available?

Reviewer #4: (No Response)

5. Is the manuscript presented in an intelligible fashion and written in standard English?

Reviewer #4: (No Response)

6. Review Comments to the Author

Reviewer #4: (No Response)

7. PLOS authors have the option to publish the peer review history of their article (what does this mean?). If published, this will include your full peer review and any attached files.

Reviewer #4: No

---

## [Editor Report · Acceptance letter]

6 Apr 2021

PONE-D-20-20388R5 

Exploring the genomic resources of seven domestic Bactrian camel populations in China through restriction site-associated DNA sequencing 

Dear Dr. Ren:

I'm pleased to inform you that your manuscript has been deemed suitable for publication in PLOS ONE. Congratulations! Your manuscript is now with our production department. 

Kind regards, 

on behalf of

Dr. Tzen-Yuh Chiang 

Academic Editor

PLOS ONE